# IMPORTANCE SAMPLING FOR MULTI-NEGATIVE MULTIMODAL DIRECT PREFERENCE OPTIMIZATION

**Xintong Li**[1]\*,**Chuhan Wang**[1]\***, Junda Wu**[1]**, Rohan Surana**[1]**, Tong Yu**[2]**,**
**Julian McAuley**[1] **, Jingbo Shang**[1]
[1]University of California, San Diego    [2]Adobe Research
{xil240,chw136,juw069,rsurana,jmcauley,jshang}@ucsd.edu
tyu@adobe.com

## ABSTRACT

Direct Preference Optimization (DPO) has recently been extended from text-only models to vision-language models. However, existing methods rely on oversimplified pairwise comparisons, generating a single negative image via basic perturbations or similarity-based retrieval, which fail to capture the complex nature of multimodal preferences, inducing optimization bias and hallucinations. To address this issue, we propose MISP-DPO, the first framework to incorporate *multiple*, semantically *diverse* negative images in multimodal DPO via the Plackett-Luce model. Our method embeds prompts and candidate images in CLIP (Contrastive Language–Image Pre-training) space and applies a sparse autoencoder to uncover semantic deviations into interpretable factors. Negative samples are selected based on reconstruction difficulty, semantic deviation from the positive, and mutual diversity, yielding broader and more informative supervision. To handle multi-negative comparisons, we adopt a Plackett–Luce objective and introduce an importance sampling strategy that improves training efficiency. Experiments across five diverse benchmarks demonstrate that MISP-DPO consistently improves multimodal alignment over prior methods, validating the effectiveness of semantic-aware, multi-negative sampling in preference-based learning.

## 1 INTRODUCTION

Direct Preference Optimization (DPO) (Rafailov et al., 2023; Amini et al., 2024) has shown great promise for aligning language models by learning from pairwise comparisons, bypassing the need for explicit reward modeling. Recent efforts have extended DPO to multimodal contexts, enhancing vision-language model (VLM) alignment through image-text feedback(Wang et al., 2024a; Jiang et al., 2024; Deng et al., 2024; Fu et al., 2025; Liu et al., 2025a; Wu et al., 2025; Xing et al., 2025). While these approaches improve multimodal alignment, they also expose new challenges, most notably persistent hallucinations and spurious correlations (Wang et al., 2024a; Fu et al., 2025; Wu et al., 2025).

A key limitation of existing multimodal DPO methods lies in how negative examples are constructed. Most approaches generate only a single negative image per comparison, typically via adversarial cropping, random perturbations, or similarity-based retrieval (Liu et al., 2025a; Fu et al., 2025; Wu et al., 2025; Xing et al., 2025). This reduces supervision to a one-dimensional deviation from the positive image, severely underspecifying the space of visual errors. For instance, as illustrated in Figure 1, avoiding a single negative depicting a "green apple" might teach the model to reject green hues but ignore mismatched contexts like "kitchen counter" or incorrect objects like "pear." As a result, optimization occurs along a narrow deviation axis, encouraging spurious shortcuts and limiting generalization across orthogonal failure modes. Even recent listwise extensions suffer from a similar issue. Although multiple negatives are introduced, they are typically generated by perturbing a single positive image, resulting in homologous variants with substantial semantic overlap (Zadeh et al., 2025; Liu et al., 2025d). The learning signal remains concentrated around closely related deviations, rather than spanning diverse semantic discrepancies (Liu et al., 2025b;c). This oversimplifies

---

\*These authors contributed equally to this work.

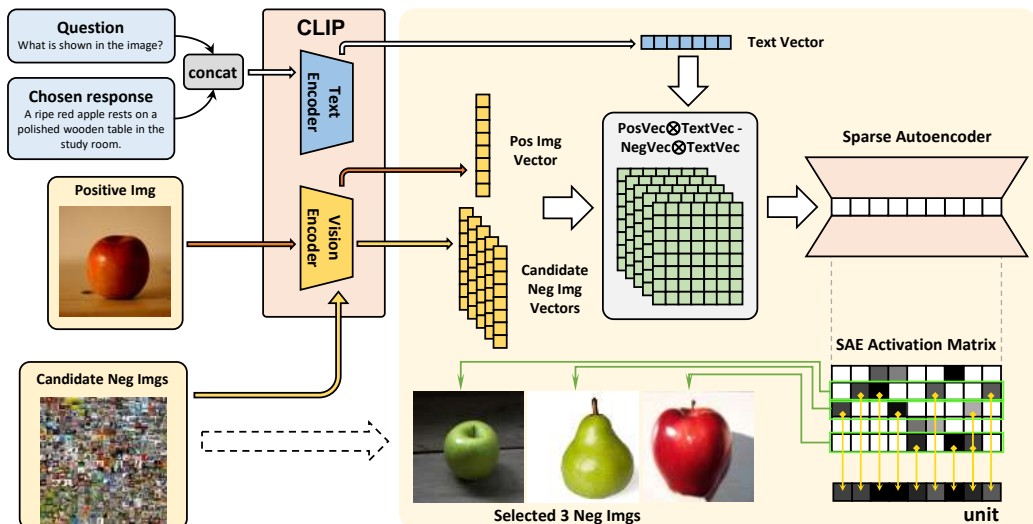

Figure 1: Overview of the MISP-DPO framework, which integrates CLIP encoding and sparse autoencoder–guided selection to identify diverse negatives for multi-negative preference optimization.

the rich space of visual inconsistencies and constrains the model's ability to disentangle distinct sources of hallucination.

The core challenge is that images lack explicit, compositional units like text tokens, making it difficult to isolate meaningful visual deviations (Sahin et al., 2024; Zeng et al., 2024; Zheng et al., 2024; Hsieh et al., 2023; Kamath et al., 2024; 2023). Naive perturbations often destroy overall coherence without isolating meaningful deviations, making it difficult to systematically explore the negative factors of model weaknesses. Effective learning requires disentangling multiple latent error factors, such as object identity, color, spatial layout, or contextual mismatch, while preserving prompt relevance. Existing methods implicitly entangle these factors within a single negative, leaving models blind to orthogonal dimensions of failure.

To address this, we propose MISP-DPO, a framework that introduces *multi*-negative, semantically *diverse* supervision into multimodal DPO. We tackle the underexplored problem of constructing multifaceted visual negatives that capture distinct semantic deviations, enabling richer and more structured preference signals. Our approach consists of two stages,

- In the first stage, we select diverse image-side negatives from a large open-domain pool. Prompts and candidate images are embedded in CLIP (Radford et al., 2021) space, and a sparse autoencoder (SAE) decomposes their semantic differences into disentangled latent factors (e.g., object, color, layout). We prioritize negatives based on: (1) reconstruction error (informativeness), (2) semantic deviation from the positive sample, and (3) mutual diversity, ensuring broad coverage of negative types.
- In the second stage, we integrate these multiple negatives into a generalized DPO objective using the Plackett–Luce model. Rather than relying on binary comparisons, our approach ranks a positive image above a diverse set of negatives, forcing the model to resolve multiple constraints simultaneously. We further introduce an importance sampling scheme guided by SAE-derived scores, improving training efficiency.

We evaluate MISP-DPO on five multimodal benchmarks (Sun et al., 2023; Guan et al., 2024; Lu et al., 2024; Tong et al., 2024; Li et al., 2023a) focused on hallucination reduction and visual grounding. Our method consistently outperforms strong baselines, achieving notable hallucination reduction and improved alignment, including a 30.09% average improvement over LLaVA-v1.5-7B (Liu et al., 2023).

Our contributions are as follows:

- We introduce a multi-negative multimodal DPO framework that provides semantically diverse supervision over orthogonal visual deviations, moving beyond single-axis negative optimization.

- We propose an efficient negative sampling strategy based on CLIP embeddings and SAE–guided importance sampling, providing informative and diverse visual discrepancies from open-domain image pools.
- Extensive evaluations demonstrate that our method substantially reduces hallucinations and achieves robust multimodal alignment across multiple benchmarks.

## 2 RELATED WORKS

**Multimodal Direct Preference Optimization.** DPO (Rafailov et al., 2023) has become a widely adopted method for aligning LLMs with human preferences due to its simplicity and stability. However, when extended to multimodal scenarios, especially for hallucination-prone tasks, standard DPO often fails to effectively incorporate visual signals, leading models to overfit textual biases and ignore image-grounded constraints. To mitigate this, recent works have adapted DPO for multimodal hallucination reduction by incorporating visual preference supervision. mDPO (Wang et al., 2024a) introduces conditional preference learning and reward anchoring, using lightweight perturbations (e.g., cropping, diffusion) to construct visual negatives. CHiP (Fu et al., 2025) further complements this with hierarchical textual supervision and a visual contrastive loss to better align fine-grained text and image semantics. While both methods demonstrate notable gains on hallucination benchmarks, they rely on limited forms of visual augmentation, often constrained to local perturbations with narrow semantic variation. Other approaches, such as S-VCO (Wu et al., 2025) and Re-Align (Xing et al., 2025), explore counterfactual or retrieval-based visual negative generation, but at the cost of high computation and limited scalability. In this work, we follow the hallucination-centric preference optimization paradigm initiated by mDPO, and propose a scalable framework for generating informative visual negatives tailored for multimodal preference learning.

**Multi-negative Preference Optimization.** Recent works in textual and recommendation domains (Amini et al., 2024; Shi et al., 2024; Baruch et al.) have extended DPO to multi-negative settings, ranking positives above multiple negatives to enhance robustness. For example, Softmax-DPO (Chen et al., 2024a) and DMPO (Shi et al., 2024) adopt soft ranking or Plackett–Luce objectives to reduce noise sensitivity. However, such techniques remain underexplored in vision-language models, where negatives must capture subtle cross-modal semantic shifts. Inspired by findings in attribute-based recognition (Yan et al., 2023; Wang et al., 2024b) showing that compact, curated subsets can match large noisy sets, we adapt this insight to multimodal preference learning. Our framework uses a sparse autoencoder in CLIP space to select semantically diverse negatives, enabling importance-weighted ranking over multiple contrastive examples and capturing fine-grained failure modes more effectively.

## 3 PRELIMINARIES

### 3.1 MULTIMODAL DIRECT PREFERENCE OPTIMIZATION

DPO (Rafailov et al., 2023) provides a principled way to align a learned policy with human preference judgments without explicitly modeling rewards. In the RLHF framework, solving for the optimal policy $\pi^*$ under a fixed reference policy $\pi_{\text{ref}}$ yields a latent reward function

$$r(x, y) = \beta \log \frac{\pi^*(y \mid x)}{\pi_{\text{ref}}(y \mid x)} + Z(x), \tag{1}$$

where $\beta$ scales the strength of alignment and $Z(x)$ is a prompt-dependent normalizer. Substituting this into the Bradley–Terry–Luce model and dropping $Z(x)$ gives a simple training objective for a parametric policy $\pi_\theta$,

$$\mathcal{L}_{\text{DPO}}(\theta) = -\mathbb{E}_{(x, y_p, y_n) \sim D} \left[ \log \sigma \big( \beta \log \tfrac{\pi_\theta(y_p|x)}{\pi_{\text{ref}}(y_p|x)} - \beta \log \tfrac{\pi_\theta(y_n|x)}{\pi_{\text{ref}}(y_n|x)} \big) \right]. \tag{2}$$

Recent work extends DPO to vision-language models (VLMs) by incorporating visual preferences. Let $x$ denote the multimodal prompt, $m_p$ a preferred image aligned with textual response $y_p$, and $m_n$ a rejected image. The multimodal reward $r(m, x, y)$ now depends on visual grounding, with preferences modeled as,

$$p^*(y_p \succ y_n \mid m, x) = \sigma \big( r(m, x, y_p) - r(m, x, y_n) \big).$$

To ensure a fair comparison across images, we hold $y_p$ fixed and vary only the image input, the multimodal DPO loss (Wang et al., 2024a; Fu et al., 2025; Wu et al., 2025) focuses on visual discrimination,

$$\mathcal{L}_{\text{DPO}_{\text{img}}}(\theta) = -\mathbb{E}_{(m_p, m_n, x, y_p) \sim D} \left[ \log \sigma \big( \beta \log \frac{\pi_\theta(y_p | m_p, x)}{\pi_{\text{ref}}(y_p | m_p, x)} - \beta \log \frac{\pi_\theta(y_p | m_n, x)}{\pi_{\text{ref}}(y_p | m_n, x)} \big) \right]. \quad (3)$$

This formulation supports joint optimization over visual and textual inputs, enabling the policy to associate preferred images with relevant multimodal features.

### 3.2 MULTI-NEGATIVE PREFERENCE OPTIMIZATION

Multi-negative preference optimization (Chen et al., 2024a) extends the Direct Preference Optimization approach (Rafailov et al., 2023), enabling language models to be trained against several negative preferences rather than just one. Instead of using the Bradley–Terry formulation for single pairwise comparisons, this method adopts the Plackett–Luce model (Plackett, 1975; Luce et al., 1959) to score a target choice in relation to an entire set of inferior alternatives.

Given a prompt $x$, a preferred response $y_p$, and a set of $N$ non-preferred responses $\mathcal{Y}_n = \{y_n^i\}_{i=1}^N$, the Plackett–Luce probability that $y_p$ is ranked above all $y_n^i$ is

$$p^*(y_p \succ \mathcal{Y}_n \mid x) = \frac{\exp\big(r(x, y_p)\big)}{\exp\big(r(x, y_p)\big) + \sum_{i=1}^N \exp\big(r(x, y_n^i)\big)}, \quad (4)$$

where $r(x, y)$ is the latent reward function. Substituting

$$r(x, y) = \beta \log \frac{\pi_\theta(y \mid x)}{\pi_{\text{ref}}(y \mid x)} + Z(x)$$

and noting that $Z(x)$ cancels in the ratio gives

$$p^*(y_p \succ \mathcal{Y}_n \mid x) = \frac{1}{1 + \sum_{i=1}^N \exp\big(\beta \Delta_i\big)}, \quad \Delta_i = \log \frac{\pi_\theta(y_n^i \mid x)}{\pi_{\text{ref}}(y_n^i \mid x)} - \log \frac{\pi_\theta(y_p \mid x)}{\pi_{\text{ref}}(y_p \mid x)}.$$

Hence the multi-negative DPO training objective becomes

$$\mathcal{L}_{\text{MN-DPO}}(\theta) = -\mathbb{E}_{(x, y_p, \mathcal{Y}_n) \sim D} \left[ \log \sigma \big( - \log \sum_{i=1}^N \exp\big(\beta \Delta_i\big) \big) \right]. \quad (5)$$

Notably, when $N = 1$, $\mathcal{L}_{\text{MN-DPO}}$ in equation 5 reduces exactly to the single-negative DPO loss.

## 4 FRAMEWORK

We propose MISP-DPO, a framework that address the limitations of single-negative supervision by introducing multi-negative learning through two core components: (1) a diverse negative sampling strategy using sparse autoencoders to identify semantically meaningful deviations, and (2) a generalized Plackett-Luce ranking objective that integrates multiple negatives to promote robust alignment. An overview of the framework is shown in Figure 1.

### 4.1 MULTI-NEGATIVE OBJECTIVES

Due to the limitations of single-negative supervision and the inherently multi-faceted nature of visual errors, we extend multimodal DPO to a multi-negative preference optimization setting. Let $\pi_\theta$ denote the VLM policy to be optimized. Each training instance consists of a multimodal prompt $x$, a preferred image $m_p$ paired with an aligned textual response $y_p$, and a set of $N$ negative images $S_n = \{m_n^i\}_{i=1}^N$ from open-domain sources. Following the Plackett-Luce formulation from Eq. equation 4, we adapt equation 5 to visual preferences,

$$\mathcal{L}_{\text{img}}(\theta; S_n) = \log \sigma \left( - \log \sum_{i \in S_n} \exp \left( \beta \log \frac{\pi_\theta(y_p \mid x, m_n^i)}{\pi_{\text{ref}}(y_p \mid x, m_n^i)} - \beta \log \frac{\pi_\theta(y_p \mid x, m_p)}{\pi_{\text{ref}}(y_p \mid x, m_p)} \right) \right) \quad (6)$$

This extends Eq. equation 3 to multiple negatives through the softmax aggregation and encourages the model to assign higher preference scores to the correct image $m_p$ compared to all negative images in $S_n$, thereby promoting more robust visual grounding.

**Lemma 4.1 (Gradient Decomposition)** *Defining the preference advantage of each negative image and the preference distribution as*

$$a_i = \beta \left( \log \frac{\pi_\theta(y_p \mid x, m_n^i)}{\pi_{\text{ref}}(y_p \mid x, m_n^i)} - \log \frac{\pi_\theta(y_p \mid x, m_p)}{\pi_{\text{ref}}(y_p \mid x, m_p)} \right), \quad p_\theta(m_n^i \mid x, m_p, y_p) = \frac{\exp(a_i)}{\sum_{j=1}^N \exp(a_j)}.$$

*Then the gradient of equation 6 decomposes as,*

$$\nabla_\theta \mathcal{L}_{\text{img}}(\theta; \mathcal{S}_n) = \beta \sigma \left( \log \sum_{i=1}^N \exp(a_i) \right) \sum_{i=1}^N p_\theta(m_n^i \mid x, m_p, y_p) \Delta_\theta(m_n^i, m_p \mid x, y_p), \quad (7)$$

*where* $\Delta_\theta(m_n^i, m_p \mid x, y_p) = \nabla_\theta \log \pi_\theta(y_p \mid x, m_n^i) - \nabla_\theta \log \pi_\theta(y_p \mid x, m_p).$

This result shows that the gradient is a weighted combination of correction signals across the image space, offering interpretability in terms of how the model adjusts its predictions in response to each visual discrepancy.

Although Eq.( 6) and its gradient give an unbiased update, they require drawing a large set of negatives from the true $p_\theta(m_n^i \mid x, m_p, y_p)$ and computing $S(\{m_n^i\})$. In realistic image domains, neither step is tractable. To alleviate this, we introduce a learnable distribution $q_\phi(m_n \mid x, m_p, y_p)$ to sample a small candidate pool $\tilde{S}_n$. Rewriting the gradient illustrated in Lemma 4.1 as an expectation under $q_\phi$ gives a *importance-sampling* estimator,

$$\nabla_\theta \mathcal{L}_{\text{img}}(\theta; \tilde{\mathcal{S}}_n) = \beta \sigma \left( \log \sum_{i \in \tilde{\mathcal{S}}_n} \exp(a_i) \right) \sum_{i \in \tilde{\mathcal{S}}_n} \frac{\exp(a_i)}{q_\phi(m_n^i \mid x, m_p, y_p)} \Delta_\theta(m_n^i, m_p \mid x, y_p). \quad (8)$$

To ensure that this approximation remains theoretically sound, we prove in Appendix A.3 that the resulting importance-sampling gradient estimator is strictly unbiased and admits a finite-sample variance bound. In particular, its mean-square error scales as $O(1/K)$, with the constant governed by the maximum importance weight. These guarantees justify the use of a small candidate pool while preserving stable optimization.

We further promote cross-modal alignment by incorporating textual preference supervision. Following recent multimodal DPO approaches (Xie et al., 2024; Wang et al., 2024a; Ouali et al., 2024), we replace traditional text-only preferences with image-grounded negative responses $y_n$ conditioned on the same prompt and image $m_p$. The corresponding DPO loss is,

$$\mathcal{L}_{\text{text}}(\theta; \tilde{\mathcal{S}}_n) = \log \sigma \left( \beta \log \frac{\pi_\theta(y_p \mid x, m_p)}{\pi_{\text{ref}}(y_p \mid x, m_p)} - \beta \log \frac{\pi_\theta(y_n \mid x, m_p)}{\pi_{\text{ref}}(y_n \mid x, m_p)} \right). \quad (9)$$

Our final loss combines both visual and textual preference signals,

$$\mathcal{L}(\theta; \tilde{\mathcal{S}}_n) = \mathcal{L}_{\text{img}}(\theta; \tilde{\mathcal{S}}_n) + \lambda \mathcal{L}_{\text{text}}(\theta; \tilde{\mathcal{S}}_n). \quad (10)$$

where $\lambda$ balances the contributions of image-based and text-based supervision. This unified formulation supports joint alignment across modalities, improving robustness and alignment quality in VLMs.

### 4.2 IMPORTANCE SAMPLING VIA SPARSE AUTOENCODER

To address the limitations of existing methods that rely on simplistic, one-dimensional negatives, we employ SAEs to disentangle and surface semantically meaningful variations in the visual space. By providing a structured and interpretable latent representation, SAEs enable principled importance sampling over diverse negative examples—prioritizing those that capture distinct failure modes and are most informative for effective preference learning.

**Embedding and Difference Vectors.** Let $\mathcal{T} = \{(m_p, x)\}$ be the training set of positive image–prompt pairs. We use CLIP's image and text encoders, $f_v$ and $f_t$, to obtain $d$-dimensional embeddings $h_v = f_v(m_p)$ and $h_t = f_t(x)$, then fuse them via outer product and vectorization $e = \text{vec}(h_v \times h_t^\top) \in \mathbb{R}^{d^2}$. For each negative candidate $m_n^i$, we form the difference vector

$$d_i = e(m_p, x) - e(m_n^i, x).$$

---

**Algorithm 1** Greedy Diversity-Promoting Selection

---

1: **Input:** Difference vectors $\{d_i\}_{i=1}^N$, scores $\{s_i\}$, encoder $\mathcal{E}$, selection size $K$
2: $\tilde{\mathcal{S}}_n \leftarrow \emptyset$
3: **while** $|\tilde{\mathcal{S}}_n| < K$ **do**
4: $\quad i^* \leftarrow \arg\max_{i \notin \tilde{\mathcal{S}}_n} \left( s_i + \beta \min_{j \in \tilde{\mathcal{S}}_n} (1 - \cos(\mathcal{E}(d_i), \mathcal{E}(d_j))) \right)$
5: $\quad \tilde{\mathcal{S}}_n \leftarrow \tilde{\mathcal{S}}_n \cup \{i^*\}$
6: **end while**
7: **Output:** selected negatives set $\tilde{\mathcal{S}}_n$

---

**Sparse Autoencoder Training.** We train an SAE with encoder $\mathcal{E}$ and decoder $\mathcal{D}$ to decompose $d_i$ into sparse latent factors. The loss combines reconstruction fidelity and activation sparsity,

$$\mathcal{L}_{\text{SAE}} = \frac{1}{|\mathcal{T}|N} \sum_{(m_p, x) \in \mathcal{T}} \sum_{i=1}^N \left\| d_i - \mathcal{D}(\mathcal{E}(d_i)) \right\|_2^2 + \gamma \sum_{j=1}^H \text{KL}(\rho \| \hat{\rho}_j), \tag{11}$$

where $\hat{\rho}_j$ is the average activation of hidden unit $j$, $\rho \in (0, 1)$ is the target average activation, and $\gamma$ balances reconstruction against sparsity.

**Diverse Negative Selection.** We score each candidate $m_n^i$ by,

$$s_i = \frac{\left\| d_i - \mathcal{D}(\mathcal{E}(d_i)) \right\|_2^2}{\max_j \ell_j} + \frac{\left\| \mathcal{E}(d_i) \right\|_1}{\max_j v_j}, \tag{12}$$

where $\ell_j$ and $v_j$ are the reconstruction error and activation magnitude across all candidates. To choose the final top-$K$ negatives $\tilde{\mathcal{S}}_n$, we run a greedy selection that maximizes coverage of distinct error types while emphasizing hard negatives. We illustrate this algorithm in detail in Algorithm 1.

The selected set $\tilde{\mathcal{S}}_n$ is then used in the importance-sampling gradient estimator of Eq. equation 8.

## 5 EXPERIMENT

### 5.1 EXPERIMENTAL SETTINGS

**Models.** We apply MISP-DPO to three widely-used multimodal LLMs: LLaVA-1.5-7B-HF, Qwen2.5-VL-7B, and Qwen2.5-VL-3B. These models are chosen due to their open availability, competitive performance, and diverse architectural designs (Chen et al., 2024b; Zhang et al., 2024). LLaVA-1.5-7B-HF (Liu et al., 2023) integrates CLIP as the vision encoder with Vicuna-1.5-7B as the language backbone. Qwen2.5-VL-7B (Bai et al., 2025) uses a proprietary vision module and the strong Qwen2.5-7B language model. Qwen2.5-VL-3B is a lightweight 3B variant of the same architecture, providing a better balance between efficiency and capability.

**Training data.** We choose RLHF-V-Dataset (Yu et al., 2024) as our training dataset. It contains more than 5K samples, each with an image and a pair of text responses indicating preference. RLHF-V provides fine-grained, segment-level human feedback on diverse vision-language instructions, which has been shown to largely reduce hallucination while preserving informativeness. We treat the paired image as the positive sample and select 3 negative images per sample from COCO training dataset (Lin et al., 2014) using our importance sampling method, enabling effective training in our MISP-DPO framework.

**Baselines.** We compare MISP-DPO against five baselines: (1) the pretrained model without preference tuning, (2) standard DPO (Rafailov et al., 2023), which uses only a single text preference without any image-based supervision, (3) mDPO (Wang et al., 2024a) and (4) CHiP (Fu et al., 2025), both of which incorporate image preferences but rely on a single negative image per comparison, and (5) a variant of our framework that uses multi-negative image preference optimization with negatives randomly sampled from the COCO dataset. All methods are trained under the same settings for a fair comparison.

| | Benchmarks | Hallucination | | | | | | Vision-Centric | | | Total |
|---|---|---|---|---|---|---|---|---|---|---|---|
| | | MMHalBench | | HallusionBench | | | POPE | WildVision | | MMVP | avg_impr. |
| | | Score (↑) | HalRate (↓) | aA (↑) | fA (↑) | qA (↑) | Acc. (↑) | Reward (↑) | WinRate (↑) | Acc. (↑) | over BASE |
| **llava-1.5-7b-hf** | Base | 2.78 | 51.04 | 47.73 | 17.63 | 12.30 | **84.37** | -55.7 | 17.0 | 60.67 | 0% |
| | DPO | 3.29 | 37.50 | 55.62 | 22.83 | 22.63 | 83.02 | -52.7 | 18.4 | 62.66 | +21.13% |
| | mDPO | 2.99 | 49.81 | 47.32 | 20.52 | 13.19 | 83.25 | -62.1 | 14.6 | 58.33 | +0.22% |
| | CHiP | 3.13 | 34.04 | 51.95 | 17.92 | 19.78 | 82.56 | -68.4 | 12.2 | 52.33 | +5.59% |
| | Random | 3.42 | 36.46 | 55.94 | 23.69 | 22.63 | 82.61 | -51.3 | 18.3 | 60.33 | +22.23% |
| | **MISP-DPO** | **3.51** | **32.29** | **57.52** | **25.43** | **24.83** | 83.94 | **-46.4** | **20.6** | **63.00** | **+30.09%** |
| **Qwen2.5-VL-7B** | Base | 4.61 | 18.09 | 70.45 | 43.06 | 45.27 | 87.65 | **33.5** | **69.2** | 77.67 | 0% |
| | DPO | 4.92 | **11.46** | 69.92 | 42.48 | 43.51 | 87.46 | 32.8 | 68.6 | 78.00 | +3.85% |
| | mDPO | 5.01 | 14.89 | 67.40 | 41.33 | 42.20 | 87.02 | 28.5 | 66.2 | 76.33 | -1.16% |
| | CHiP | 5.02 | 13.83 | 66.14 | 39.02 | 40.88 | 88.18 | 28.5 | 66.4 | 77.00 | -1.33% |
| | Random | 4.75 | 16.67 | 70.24 | 42.48 | 43.95 | 87.60 | 30.7 | 67.8 | 78.33 | -0.36% |
| | **MISP-DPO** | **5.05** | **11.46** | **71.24** | **43.77** | **45.61** | **88.66** | 32.4 | 68.4 | **79.00** | **+5.35%** |
| **Qwen2.5-VL-3B** | Base | 4.20 | 22.34 | 64.67 | 37.57 | 36.70 | 87.48 | -0.1 | 46.6 | 70.60 | 0% |
| | DPO | 4.50 | 18.75 | 65.19 | 36.41 | 37.14 | 87.42 | 7.5 | 51.2 | 71.33 | +13.12% |
| | mDPO | 4.47 | 21.28 | 62.88 | 35.84 | 37.14 | 87.65 | 7.2 | 50.8 | 69.33 | +10.51% |
| | CHiP | 4.51 | 15.96 | 62.14 | 36.13 | 34.29 | 87.30 | 6.3 | 51.0 | 70.33 | +11.81% |
| | Random | 4.27 | 16.67 | 64.98 | **38.44** | 37.36 | 87.52 | 3.6 | 48.6 | 72.00 | +8.57% |
| | **MISP-DPO** | **4.61** | **13.54** | **65.51** | **38.44** | **38.02** | **87.77** | **8.6** | **52.4** | **74.25** | **+19.89%** |

Table 1: Comparison of MISP-DPO against baseline methods across five vision–language benchmarks and three model backbones. The benchmarks cover hallucination detection and vision-centric reasoning. Average improvement over BASE is reported. ↑: higher is better; ↓: lower is better.

**Evaluation Benchmarks.** We evaluate MISP-DPO and all baselines across five benchmarks spanning hallucination detection and vision-centric reasoning. MMHal-Bench(Sun et al., 2023) is a hallucination-focused VQA benchmark covering 8 question types and 12 object categories. HallusionBench(Guan et al., 2024) measures visual and factual hallucinations; we report all-answer accuracy (aA), figure-based accuracy (fA), and question-type accuracy (qA). POPE(Li et al., 2023b) evaluates object hallucination in VLMs via Yes/No probing under random, popular, and adversarial object settings. WildVision(Lu et al., 2024) evaluates real-world user preference alignment with 500 curated human-model interaction samples; we report reward score and win rate. MMVP(Tong et al., 2024) assesses fine-grained visual reasoning using CLIP-blind image pairs, with accuracy reported over 135 zero-shot questions across 9 pattern types. Except for MMHal-Bench, all evaluations are conducted using VLMEvalKit(Duan et al., 2024), an open-source evaluation toolkit for vision-language models. For MMHal-Bench, we use GPT-4.1-mini(Achiam et al., 2023) as the evaluator and report overall response quality and hallucination rate.

**Implementation Details.** For training the Sparse Autoencoder, we set the latent dimension to 128 and the sparsity weight $\gamma$ to 1, balancing reconstruction fidelity with latent sparsity. Following prior work on multi-negative preference optimization (Chen et al., 2024a), we select three negative images per instance. For multimodal DPO training, we set the supervision balance parameter $\lambda$ to 1 to equally weight image-based and text-based preferences. All models are fine-tuned using 2 NVIDIA A100 GPUs, with a per-device batch size of 2, gradient accumulation steps of 8 (yielding an effective batch size of 32), and a learning rate of $10^{-5}$. The preference optimization coefficient $\beta$ is set to 0.5. Following prior work on mDPO, we adopt LoRA for parameter-efficient tuning, with a rank of 64 and scaling factor $\alpha = 128$. For baseline methods, we strictly follow their original settings: mDPO is trained for 3 epochs with the same learning rate (1e-5), $\beta = 0.1$; CHiP is trained for 3 epochs with a batch size of 32, a learning rate of 5e-7, $\beta = 0.5$, and full-parameter finetuning.

## 5.2 OVERALL PERFORMANCE IMPROVEMENT

Table 1 shows the performance of MISP-DPO and baselines across five representative benchmarks, grouped into two categories: hallucination detection (MMHalBench, HallusionBench, POPE) and vision-centric reasoning (WildVision, MMVP). Our proposed MISP-DPO consistently achieves superior results over all evaluation domains and model backbones.

The largest gains appear on hallucination benchmarks. MISP-DPO substantially reduces hallucination rates on MMHalBench (e.g., 32.29%, 11.46%, and 13.54% across different backbones) while

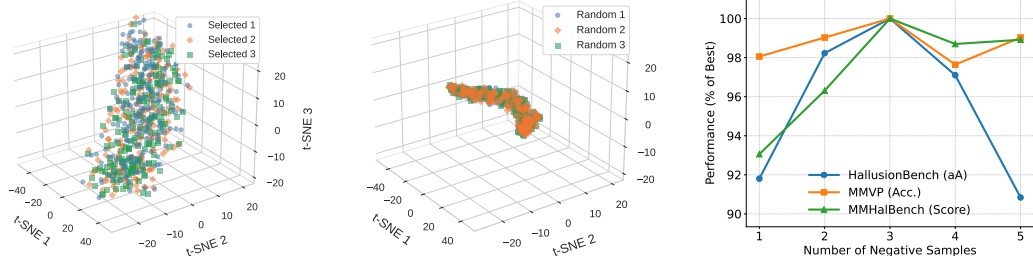

Figure 2: t-SNE visualizations and benchmark results for negative sampling. **Left:** importance-sampled negatives selected using SAE scoring exhibit broad semantic dispersion across three selections. **Middle:** randomly sampled negatives form tight, low-diversity clusters. **Right:** performance across benchmarks with different numbers of negatives selected by MISP-DPO.

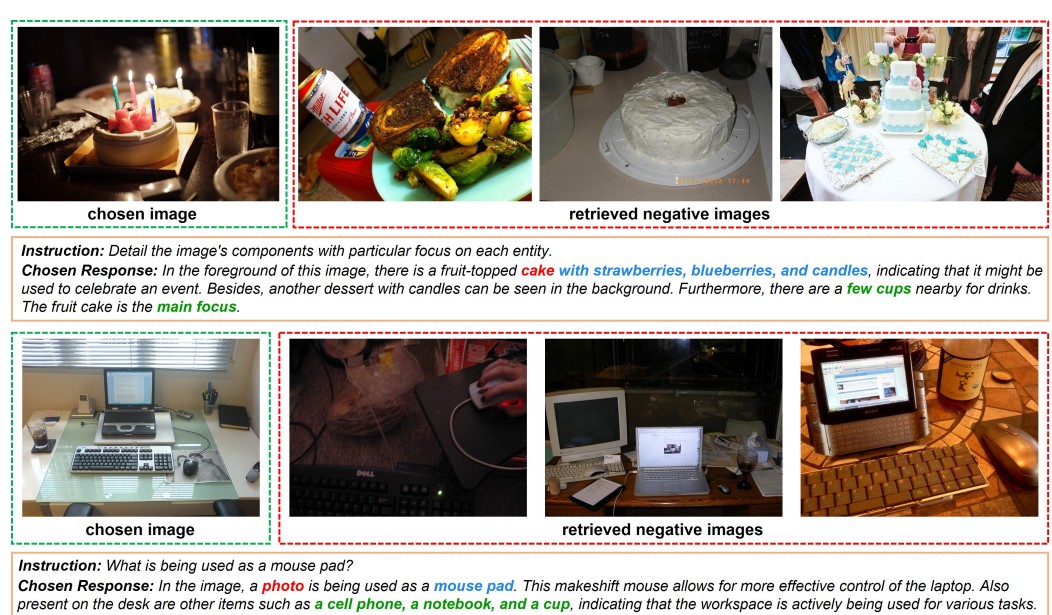

Figure 3: Negative image retrieval using our MISP-DPO method. Each row shows a chosen image and three negatives; highlighted phrases in **red**, **blue** and **green** mark mismatches with negatives.

also achieving the highest accuracy across all HallusionBench metrics. POPE further confirms these advantages. These improvements stem from the combination of diverse negative sampling, which exposes the model to varied error types such as object mismatches and attribute distortions, and importance sampling, which prioritizes hard negatives with high reconstruction errors from SAE, leading to stronger visual grounding. On vision-centric reasoning tasks, MISP-DPO also provides consistent gains. For example, it achieves the best reward (+8.6) and win rate (52.4) on WildVision for Qwen2.5-VL-3B, while also outperforming baselines on MMVP across different model sizes. These results suggest that our method not only suppresses hallucinations but also enhances the model's ability to generate fine-grained, visually aligned responses.

We also conduct experiments under different $\beta$ values on MMHalBench to balance reward learning and regularization. Figure 5 reveals performance peaks at $\beta$ ranging from 0.45 to 0.75, with degradation at extremes $\beta = 0.1/1.0$. We choose $\beta = 0.5$ for optimal trade-off between hallucination control and response quality, as it maximizes accuracy while minimizing hallucination rates across all backbones.

## 5.3 EFFECTIVENESS OF IMPORTANCE SAMPLING

We analyze the impact of our importance sampling strategy using t-SNE visualizations of high-quality negative images, shown in Figure 2. The left plot displays negatives selected by our SAE-guided strategy, while the middle shows randomly sampled ones. Importance-sampled negatives

| Model | Benchmark | MMHalBench | | HallusionBench | | | POPE | WildVision | | MMVP |
|---|---|---|---|---|---|---|---|---|---|---|
| | | Score (↑) | HalRate (↓) | aA (↑) | fA (↑) | qA (↑) | Acc. (↑) | Reward (↑) | WinRate (↑) | Acc. (↑) |
| **Qwen2.5-7B** | mDPO | 5.01 | 14.89 | 67.40 | 41.33 | 42.20 | 87.02 | 28.5 | 66.2 | 76.33 |
| | diffusion | **5.12** | 12.50 | 69.50 | 43.64 | 43.95 | 87.52 | 30.7 | 66.4 | 78.00 |
| | crop+diffusion | 4.92 | 13.54 | 70.35 | 43.35 | 44.61 | 87.35 | 30.7 | 66.4 | 78.33 |
| | similarity | 5.00 | 12.50 | 69.82 | 42.77 | 44.17 | 87.24 | 30.1 | 66.2 | 77.67 |
| | **MISP-DPO** | 5.05 | **11.46** | **71.24** | **43.77** | **45.61** | **88.66** | **32.4** | **68.4** | **79.00** |
| **Qwen2.5-3B** | mDPO | 4.47 | 21.28 | 62.88 | 35.84 | 37.14 | 87.65 | 7.2 | 50.8 | 69.33 |
| | diffusion | 4.56 | 14.58 | 65.19 | **39.31** | 37.14 | 87.35 | 8.0 | **52.4** | 71.33 |
| | crop+diffusion | 4.39 | 17.70 | 64.03 | 38.44 | 36.48 | 87.20 | 5.3 | 49.8 | 71.33 |
| | similarity | 4.20 | 19.79 | 65.08 | 39.01 | 36.92 | 87.62 | 4.7 | 49.4 | 70.30 |
| | **MISP-DPO** | **4.61** | **13.54** | **65.51** | 38.44 | **38.02** | **87.77** | **8.6** | **52.4** | **72.00** |

Table 2: Comparison of different negative sampling strategies across five benchmarks. All variants share the same model and loss design, differing only in how negative images are constructed.

are well-dispersed across the embedding space, indicating high semantic diversity, whereas random negatives cluster tightly, indicating redundancy and limited coverage. This distribution reveals that our method captures heterogeneous error modes rather than collapsing onto a narrow, repetitive subset of examples, thereby providing more informative supervision.

As illustrated in Figure 3, our selected negatives for the fruit cake image exhibit clear semantic deviations: one introduces different types of cakes with distinct decorations, while others include unrelated food items such as sandwiches, or scenes where the cake is present but not the main visual focus, thereby weakening semantic alignment. Similarly, in the workspace scene, negative examples capture meaningful variations such as differences in desk arrangement and surrounding objects—including notebooks and cups—each impacting multimodal alignment differently. These examples highlight how our sampling method uncovers diverse error modes, encouraging the model to learn more robust visual distinctions, strengthening training signals, and improving generalization beyond narrow, one-dimensional deviations.

### 5.4 COMPARISON OF NEGATIVE SAMPLING STRATEGIES

We evaluate five negative sampling strategies that share the same model architecture and loss formulation but differ in how negative images are constructed: (1) mDPO, which relies on a single crop-generated negative; (2) diffusion, which combines one diffusion negative with two negatives selected by our method; (3) crop+diffusion, which mixes one cropped, one diffusion, and one MISP-DPO-selected negative; (4) similarity, where three negatives are retrieved based on similarity to the positive image; and (5) our model. As shown in Table 2, mDPO yields the weakest performance, highlighting the limitations of single-negative supervision. Among multi-negative variants, the diffusion strategy outperforms crop+diffusion, suggesting that a higher proportion of semantically diverse negatives from our method improves supervision quality. The similarity variant performs worse than diffusion, underscoring that naive retrieval of visually similar negatives does not provide the challenging guidance needed. In contrast, MISP-DPO consistently achieves the best scores across all benchmarks, validating the effectiveness of structured multi-negative selection via SAE-guided importance sampling. Additional results are reported in Table 3 in the appendix.

Beyond the choice of sampling strategy, we further explore the number of negatives required. Figure 2 (right) shows performance across HallusionBench, MMVP, and MMHalBench with varying numbers of negatives. One or two negatives are insufficient to provide high-quality supervision. Performance peaks at three negatives, while increasing beyond this offers no further benefit. For HallusionBench, adding five negatives even reduces performance, likely due to noise introduced by redundant or low-quality samples. Together, these results demonstrate that three carefully chosen negatives strike the best balance between informativeness and robustness.

### 5.5 HALLUCINATION REDUCTION WITH MISP-DPO.

Figure 6 in the Appendix presents qualitative comparisons highlighting the impact of multi-negative supervision. Baseline methods such as DPO and CHiP frequently introduce hallucinated details

| Method | HallusionBench | | | POPE | MMVP |
|---|---|---|---|---|---|
| | aA(↑) | fA(↑) | qA(↑) | Acc.(↑) | Acc.(↑) |
| w/o PL loss | 64.34 | 36.64 | 37.05 | 87.37 | 72.33 |
| w/o SAE neg | 64.98 | **38.44** | 37.36 | 87.52 | 72.00 |
| Isolate Text | 63.45 | 34.08 | 35.27 | 85.43 | 71.66 |
| Isolate Image | 61.59 | 32.11 | 35.44 | 85.01 | 71.30 |
| **MISP-DPO** | **65.51** | **38.44** | **38.02** | **87.77** | **74.25** |

**(a) Component Ablation**

| Method | LLaVA | Qwen-7B | Qwen-3B |
|---|---|---|---|
| Base | 64.05 | 77.13 | 73.99 |
| DPO | 65.07 | **77.24** | 74.02 |
| mDPO | 65.35 | 75.53 | 73.20 |
| CHiP | 60.98 | 76.78 | 74.10 |
| Random | 65.91 | 77.16 | 74.25 |
| **MISP-DPO** | **66.26** | 77.18 | **74.32** |

**(b) SEED-Bench Generalization**

Figure 4: Left: Component ablation on Qwen2.5-VL-3B. Right: SEED-Bench evaluation across three backbones. MISP-DPO improves both hallucination-oriented and general multimodal reasoning benchmarks.

(e.g., incorrect objects, colors, or spatial relations), while MISP-DPO generates more faithful and grounded descriptions. For instance, in the first example, only MISP-DPO correctly identifies that the image lacks sand and accurately describes the bird. These results illustrate that incorporating diverse negatives enables the model to better distinguish relevant from spurious cues, improving factual accuracy in vision-language alignment.

## 5.6 CONTRIBUTION OF EACH COMPONENT IN MISP-DPO

We perform controlled ablations on Qwen2.5-VL-3B to isolate the contribution of each core component in Figure 4 (a). Replacing the Plackett–Luce multi-negative objective with a simple averaged pairwise loss consistently degrades performance, indicating that jointly comparing a positive against multiple negatives provides stronger and more structured supervision than independent pairwise comparisons. Substituting SAE-guided negatives with randomly sampled images also reduces accuracy, confirming that semantically informative and diverse negatives are crucial for effective preference learning. Finally, training with image-only or text-only loss results in the largest performance drops, highlighting the necessity of complementary multimodal signals for robust visual grounding. Overall, MISP-DPO achieves the best results, indicating that its gains are derived from the synergy between structured multi-negative construction and principled ranking optimization.

## 5.7 GENERALIZATION BEYOND HALLUCINATION BENCHMARKS

To assess whether the improvements of MISP-DPO extend beyond hallucination-focused tasks, we further evaluate on SEED-Bench (Li et al., 2023a), a general-purpose multimodal benchmark covering reasoning, visual understanding, and instruction following. As shown in Figure 4 (b), MISP-DPO consistently matches or outperforms all baselines across three backbones. Notably, the gains over random multi-negative sampling indicate that improvements stem from semantically structured negative construction and principled multi-negative preference optimization instead of increased supervision alone. These results demonstrate that our method improves the overall multimodal reasoning capability rather than overfitting to hallucination-specific metrics.

## 6 CONCLUSION

We present MISP-DPO, a novel framework that introduces multi-negative, semantically diverse supervision into multimodal Direct Preference Optimization. By leveraging CLIP-based embeddings and a sparse autoencoder, our method efficiently selects image-side negatives that vary across multiple semantic facets and reflect diverse failure modes. These negatives are integrated into a Plackett-Luce-style ranking objective with importance sampling, enabling the model to learn from richer and more structured supervision. The method remains efficient and scalable for real-world multimodal applications. Extensive experiments across five benchmarks demonstrate that MISP-DPO consistently outperforms strong baselines in hallucination reduction and visual grounding. While our evaluations rely on GPT-based scoring—which may introduce bias or inconsistency when assessing fine-grained alignment—our findings validate the effectiveness of semantic-aware, multi-negative sampling for robust multimodal alignment and open up promising directions for scalable and interpretable preference-based learning.

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

## A APPENDIX

### A.1 EFFECT OF $\beta$ ON HALLUCINATION AND QUALITY

Figure 5 illustrates the impact of different $\beta$ values on both response quality (left) and hallucination rate (right) on MMHalBench. We observe that the model performs best when $\beta$ lies in the range of 0.45 to 0.75, achieving a good balance between response quality and hallucination suppression. To ensure both accuracy and stability, we set $\beta = 0.5$ as the default value in all main experiments.

### A.2 EXAMPLES OF HALLUCINATION REDUCTION WITH MISP-DPO

To illustrate the impact of our approach, Figure 6 presents qualitative comparisons across three representative prompts, highlighting the improvements brought by MISP-DPO over baselines including LLaVA, DPO, and CHiP. MISP-DPO demonstrates a stronger ability to avoid hallucinations and produce factually accurate descriptions grounded in visual evidence. In the first example, it correctly identifies the absence of sand and avoids misidentifying the bird species. In the second case, it faithfully describes the structure and positioning of the gloves despite occlusion. In the final example, it provides a precise spatial interpretation of the two watches without fabricating brand-specific details. These results suggest that our multi-negative supervision strategy improves the model's sensitivity to fine-grained semantic cues and its ability to reject spurious correlations and hallucinated attributes, leading to more factually consistent vision-language generation.

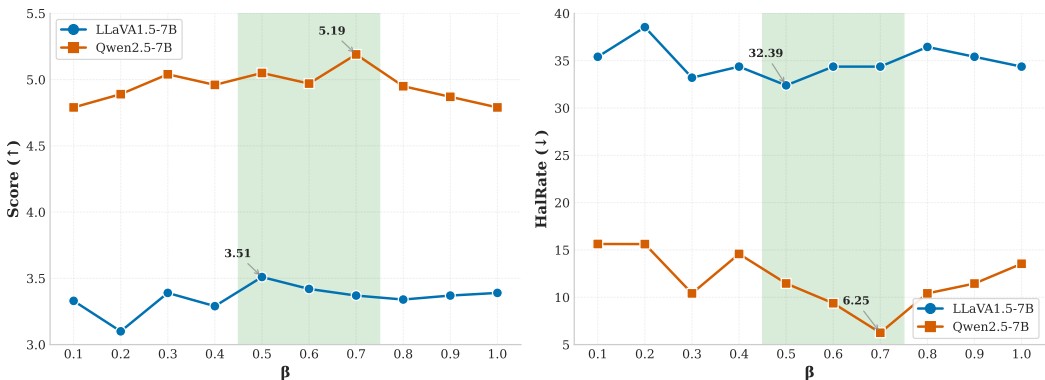

Figure 5: Performance comparison of Score and Hallucination Rate across different $\beta$ values for on MMHalBench.

### A.3 UNBIASEDNESS AND FINITE-SAMPLE VARIANCE BOUNDS FOR IMPORTANCE SAMPLING GRADIENT

For convenience, we define the inner gradient term in Lemma 4.1,

$$g(\theta) := \sum_{i=1}^{N} p_\theta(m_n^i \mid x, m_p, y_p) \, \Delta_\theta(m_n^i, m_p \mid x, y_p). \tag{13}$$

Our importance-sampling scheme (Sec. 4.1) introduces a proposal distribution $q_\phi(m_n \mid x, m_p, y_p)$ induced by the CLIP+SAE pipeline, and samples a small candidate pool $\widetilde{\mathcal{S}}_n = \{m_n^k\}_{k=1}^K$ from $q_\phi$. We then approximate the exact gradient using an importance-weighted estimator.

#### A.3.1 UNBIASEDNESS OF THE IS GRADIENT

Let the importance weight be

$$w(m_n) = \frac{p_\theta(m_n \mid x, m_p, y_p)}{q_\phi(m_n \mid x, m_p, y_p)}.$$

**Lemma A.1 (Unbiased IS gradient)** *Assume that $q_\phi(m_n \mid x, m_p, y_p) > 0$ whenever $p_\theta(m_n \mid x, m_p, y_p) > 0$. Define the Monte Carlo estimator of $g(\theta)$ as,*

$$\widehat{g}_k(\theta) := \frac{1}{K} \sum_{k=1}^{K} w\big(m_n^k\big) \, \Delta_\theta\big(m_n^k, m_p \mid x, y_p\big), \quad m_n^k \sim q_\phi(\cdot \mid x, m_p, y_p).$$

*Then $\hat{g}_k(\theta)$ is an unbiased estimator of $g(\theta)$,*

$$\mathbb{E}_{\{m_n^k\} \sim q_\phi}\big[\widehat{g}_k(\theta)\big] = g(\theta). \tag{14}$$

**Proof A.1** *Using linearity of expectation and the definition of importance weights,*

$$\begin{aligned}
\mathbb{E}_{\{m_n^k\} \sim q_\phi}\big[\widehat{g}_k(\theta)\big] &= \mathbb{E}_{m \sim q_\phi}\big[w(m_n) \, \Delta_\theta(m_n, m_p \mid x, y_p)\big] \\
&= \sum_{i=1}^{N} q_\phi(m_n^i \mid x, m_p, y_p) \frac{p_\theta(m_n^i \mid x, m_p, y_p)}{q_\phi(m_n^i \mid x, m_p, y_p)} \, \Delta_\theta(m_n^i, m_p \mid x, y_p) \\
&= \sum_{i=1}^{N} p_\theta(m_n^i \mid x, m_p, y_p) \, \Delta_\theta(m_n^i, m_p \mid x, y_p) \\
&= g(\theta),
\end{aligned}$$

*which proves the claim.*

Thus, the IS estimator recovers the exact gradient term in expectation, and hence the overall gradient in Eq. 7,

$$\nabla_\theta L_{\text{img}}(\theta; \mathcal{S}_n) = \beta \, \sigma\Big( \log \sum_{i=1}^{N} \exp(a_i) \Big) g(\theta), \tag{15}$$

admits an unbiased importance-sampling estimator by replacing $g(\theta)$ with $\widehat{g}_k(\theta)$.

### A.3.2 VARIANCE BOUND UNDER FINITE NEGATIVE SAMPLING

We next calculate the variance of $\widehat{g}_k(\theta)$ when only a finite number $K$ of negatives are sampled.

Define the maximum importance weight,

$$\|w\|_\infty := \max_{m_n \in \mathcal{S}_n} \frac{p_\theta(m_n \mid x, m_p, y_p)}{q_\phi(m_n \mid x, m_p, y_p)}.$$

Assume that the gradient differences are bounded, which in practice is achieved by gradient clipping [4-6]. There exists $L < \infty$ such that,

$$\big\|\Delta_\theta(m_n, m_p \mid x, y_p)\big\|_2 \leq L \quad \forall m_n \in \mathcal{S}_n. \tag{16}$$

**Lemma A.2 (Variance bound)** *Under the assumptions above, the mean-square error of the IS gradient estimator isbounded as,*

$$\mathbb{E}\big[\|\widehat{g}_k(\theta) - g(\theta)\|_2^2\big] \leq \frac{L^2}{K} \|w\|_\infty.$$

*Consequently, the expected absolute deviation satisfies,*

$$\mathbb{E}\big[\|\widehat{g}_k(\theta) - g(\theta)\|_2\big] \leq L \sqrt{\frac{\|w\|_\infty}{K}}. \tag{17}$$

**Proof A.2** *Because the $K$ samples are i.i.d.,*

$$\text{Var}\big(\widehat{g}_k(\theta)\big) = \frac{1}{K} \, \text{Var}\big(w(m_n)\Delta_\theta(m_n, m_p \mid x, y_p)\big).$$

*Using the bound $\left\|\Delta_\theta(m_n, m_p \mid x, y_p)\right\|_2 \le L$, we obtain*

$$\mathbb{E}\left[\|\widehat{g}_k(\theta) - g(\theta)\|_2^2\right] \le \frac{1}{K}\,\mathbb{E}_{m_n \sim q_\phi}\left[\|w(m_n)\Delta_\theta(m_n, m_p \mid x, y_p)\|_2^2\right]$$
$$\le \frac{L^2}{K}\,\mathbb{E}_{m_n \sim q_\phi}\left[w(m_n)^2\right].$$

*Moreover,*

$$\mathbb{E}_{m_n \sim q_\phi}\left[w(m_n)^2\right] \le \|w\|_\infty\,\mathbb{E}_{m \sim q_\phi}[w(m_n)]$$
$$= \|w\|_\infty \sum_{m_n} q_\phi(m_n)\frac{p_\theta(m_n)}{q_\phi(m_n)} = \|w\|_\infty \sum_{m_n} p_\theta(m_n) = \|w\|_\infty. \quad (18)$$

*Combining these yields the desired variance bound. Applying Cauchy–Schwarz gives the final variance bound.*

In our setting, the SAE-guided sampler is explicitly designed to approximate the target distribution with diverse but not overly peaky weights, which empirically keeps $\|w\|_\infty$ small and leads to stable optimization.

## A.4 NEGATIVE SAMPLING STRATEGIES ON LLaVA-1.5-7B MODEL

Table 3 shows that LLaVA-1.5-7B follows the same trend as Qwen2.5. The single-negative baseline (mDPO) performs the weakest, while adding more negatives through diffusion or crop+diffusion gives only moderate improvements.

In contrast, MISP-DPO consistently achieves the best results, notably lowering hallucination rate and improving accuracy across HallusionBench, POPE, WildVision, and MMVP. This confirms that our SAE-guided multi-negative sampling generalizes across model families, providing stronger guidance even for smaller-scale vision–language models.

| Model | Benchmark | MMHalBench | | HallusionBench | | | POPE | WildVision | | MMVP |
|---|---|---|---|---|---|---|---|---|---|---|
| | | Score (↑) | HalRate (↓) | aA (↑) | fA (↑) | qA (↑) | Acc. (↑) | Reward (↑) | WinRate (↑) | Acc. (↑) |
| | mDPO | 2.99 | 49.81 | 47.32 | 20.52 | 13.19 | 83.25 | -62.1 | 14.6 | 58.33 |
| **llava-1.5-7b** | diffusion | 3.49 | 33.33 | 52.57 | 21.38 | 18.02 | 83.25 | −53.0 | 18.8 | 61.33 |
| | crop+diffusion | 3.44 | 35.42 | 53.63 | 22.54 | 19.78 | 83.80 | -50.5 | 20.0 | 61.00 |
| | **MISP-DPO** | **3.51** | **32.29** | **57.52** | **25.43** | **24.83** | **83.94** | **-46.4** | **20.6** | **63.00** |

Table 3: llava performance

## A.5 SELECTION COST AND SCALABILITY

Our SAE-based negative-selection framework is designed to be computationally efficient and lightweight. Using a 5.7k RLHF-V dataset together with 118k COCO images on a single A100 GPU, the computation of CLIP embeddings takes 4 minutes, SAE training 9 minutes, and the negative selection stage 20 minutes, resulting in a total runtime of 33 minutes. This corresponds to only **0.37 seconds per positive sample** to select three semantically informative negatives, demonstrating the practicality and efficiency of our importance-sampling strategy.

During preference training, incorporating three negatives introduces a modest and predictable overhead: MISP-DPO runs at approximately 3.6s per sample, compared to 2.6s for DPO and 3.1s for mDPO, with CHiP exhibiting similar runtime. The overall pipeline scales linearly with dataset size and remains stable under the Plackett–Luce objective and importance-sampling estimator.

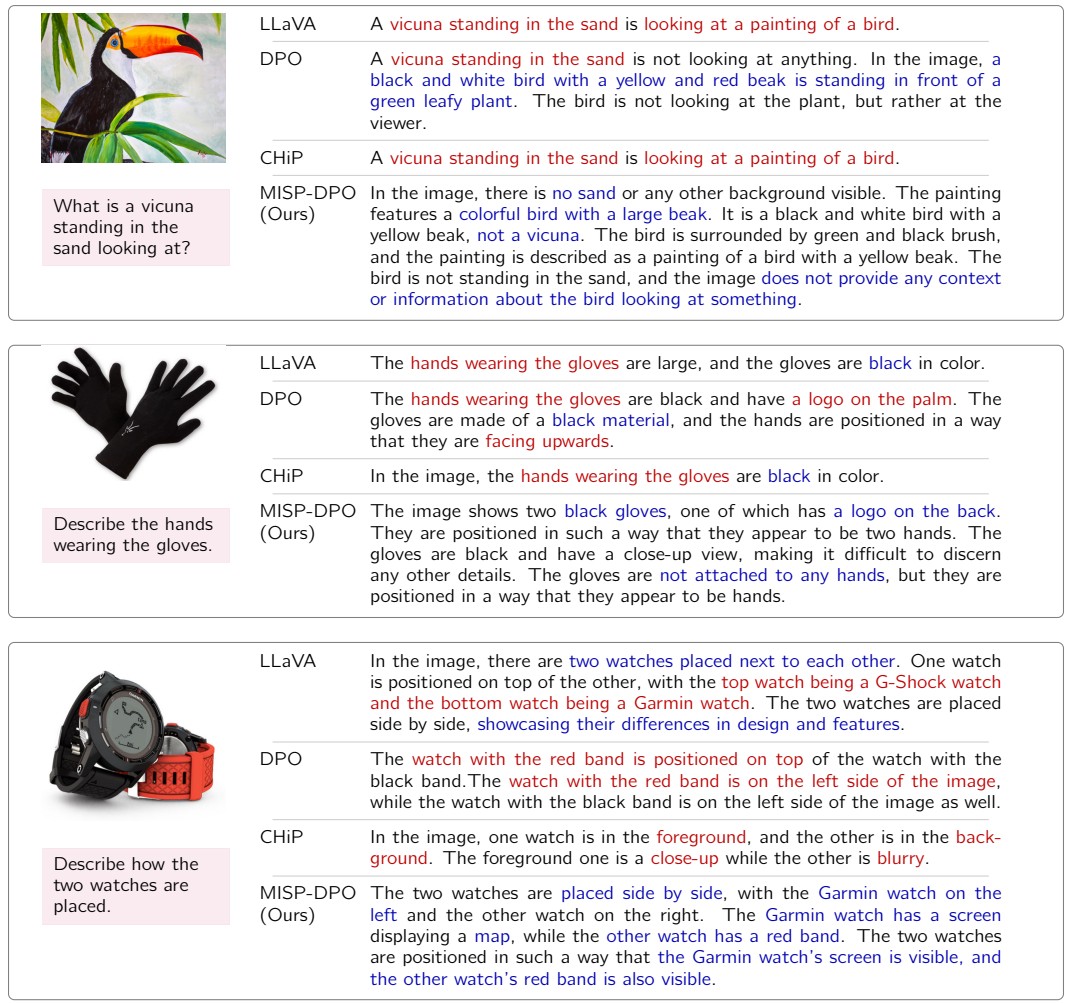

Figure 6: Comparison of responses from LLaVA model training through different methods: pretrained, DPO, CHiP, and MISP-DPO. Blue text indicates faithful descriptions; red text marks hallucinated or unsupported content.

