# OpenReview forum: "Importance Sampling for Multi-Negative Multimodal Direct Preference Optimization"
_ICLR.cc/2026/Conference — ICLR 2026 Poster_

### Official Review · Reviewer_h9st · 2025-10-29

**Soundness:** 2
**Presentation:** 3
**Contribution:** 2
**Rating:** 6
**Confidence:** 4

**Summary:**

The paper presents MISP-DPO, a framework for multimodal Direct Preference Optimization that uses multiple semantically diverse negative samples instead of a single one. It combines a Plackett–Luce ranking objective with importance sampling guided by a Sparse Autoencoder trained in CLIP space to select informative negatives. Experiments on several benchmarks show improved multimodal alignment and reduced hallucination compared to existing DPO methods.

**Strengths:**

The paper clearly identifies a weakness in current multimodal DPO frameworks—the oversimplified single-negative setup—and proposes a principled multi-negative formulation to address it.

Experimental evaluation is extensive, including comparisons across multiple models and benchmarks, with consistent quantitative gains in hallucination reduction.

**Weaknesses:**

The novelty may be moderate: it mainly leverages existing models to extract multiple negative samples, without introducing substantial theoretical or methodological innovation.

The paper does not deeply analyze computational overhead or training stability when incorporating multiple negatives, which could affect scalability for larger datasets.

It remains unclear whether the improvements generalize beyond hallucination-oriented tasks (e.g., to reasoning or instruction following).

**Questions:**

1. How does the proposed multi-negative sampling strategy affect training efficiency and scalability when applied to larger datasets?

2. Could the authors provide a more detailed analysis of computational overhead introduced by the sparse autoencoder and importance sampling modules?

3. Beyond hallucination reduction, has the method been evaluated on reasoning or instruction-following tasks to assess generalization across multimodal objectives?

I will adjust my score based on the authors’ response.

---

> ### Author Response · Authors · 2025-11-22
> **Response to Reviewer h9st [1/3]**
>
> We thank the reviewer for their positive assessment and thoughtful feedback. We appreciate the recognition of our contributions and address the concerns raised below.
>
> **Response to Weakness 1**
>
> We thank the reviewer for the comment. We would like to clarify that our contribution is not limited to “leveraging existing models to extract multiple negatives.” The novelty lies in how we leverage them. We are the first to introduce an SAE-guided importance sampling framework for constructing multi-facet, semantically diverse image negatives for multimodal DPO. This is a new formulation because it disentangles semantic deviation factors from CLIP’s cross-modal embedding and uses them to select negatives along multiple orthogonal visual dimensions (object, layout, attributes, context). Prior multimodal DPO methods either use a single negative or generate listwise negatives by perturbing one image, yielding only homologous variants. None provides a mechanism for discovering open-domain, multi-facet visual discrepancies.
>
> Our framework, therefore, introduces three coupled innovations:
> - SAE-guided semantic decomposition to surface diverse visual deviations.
> - Importance sampling to obtain a compact but informative negative set without full dataset scanning.
> - A multimodal Plackett–Luce objective that integrates these negatives into an effective multi-negative DPO framework.
>
> To strengthen our contribution, we have added a theoretical variance and bias guarantee for our importance-sampling estimator. All detailed analyses, statements, and proofs are provided in the appendix.
>
> **Unbiased Importance Sampling Gradient**
>
> Define the Monte Carlo estimator of $g(\theta)$ as,
> \begin{equation}
>     \hat{g_k(\theta)} = \frac{1}{K} \sum_{k=1}^K
>     w\big(m_n^k \big) \Delta_\theta\big(m_n^k, m_p \mid x, y_p\big),\quad
>     m_n^k \sim q_\phi(\cdot \mid x, m_p, y_p).
> \end{equation}
>
> Then $\hat{g_k(\theta)}$ is an unbiased estimator of $g(\theta)$,
> \begin{equation}
>    E_{\{m_n^k\}\sim q_\phi}
>     \big[ \hat{g_k(\theta)} \big]
>     = g(\theta).
> \end{equation}
>
> **Variance Bound Under Finite Negative Sampling**
>
> Define the maximum importance weight, $ \|w_\infty\|
>     := \max_{m_n \in \mathcal{S_n}}
>     \frac{p_\theta(m_n \mid x, m_p, y_p)}{q_\phi(m_n \mid x, m_p, y_p)}$.
> There exists $L < \infty$ such that,
> \begin{equation}
>     \big\|\Delta_\theta(m_n, m_p \mid x, y_p)\big\|_2 \le L
>     \quad \forall m_n \in \mathcal{S_n}.
> \end{equation}
> The mean-square error of the IS gradient estimator is bounded as,
> \begin{equation}
>     E\big[\|\hat{g}_k(\theta) - g(\theta)\|_2^2\big] \le  \frac{L^2}{K} |w|.
> \end{equation}
> where $K$ is the number of sampled negatives. Consequently, the expected absolute deviation satisfies,
> \begin{equation}
>     E\big[\|\hat{g}_k(\theta) - g(\theta)\|_2\big] \le  L\sqrt{\frac{\|w\|}{K}},
> \end{equation}
>
> where $\|w\|$ represents $\|w\|_{\infty}$.
>
> In our setting, the SAE-guided sampler is explicitly designed to approximate the target distribution with diverse but not overly peaky weights, which empirically keeps $\|w\|_\infty$ small and leads to stable optimization.

---

> ### Author Response · Authors · 2025-11-22
> **Response to Reviewer h9st [2/3]**
>
> **Response to Weakness 2 and Question 1**
>
> Our multi-negative sampling strategy is designed to remain efficient and scalable even on larger datasets. The main potential overhead lies in negative construction, which would be expensive if one attempted to score all 118k COCO images. This is precisely why we introduce SAE-guided importance sampling. Using only CLIP features and a lightweight 1-layer SAE, we select three high-quality, multi-facet negatives without any exhaustive retrieval. The entire negative construction process (SAE training + per-image selection) finishes in 33 minutes on a single A100 (equivalent to only ~0.37 seconds to select three negatives per positive example), and scales linearly with dataset size, since each image is processed once with no quadratic or combinatorial operations.
>
> In terms of training cost, using three negative images naturally introduces a small overhead compared with baselines that use a single negative. Text-only DPO trains in ~2.6s per sample, mDPO (with a single perturbed negative) takes ~3.1s, and CHiP exhibits a similar runtime to ours. Under the same setting, our framework completes in ~3.6s, which is a modest and predictable increase given the richer supervision and yields significantly higher performance. Importantly, we did not observe any training instability. Both the PL objective and the IS estimator remain well-behaved throughout all experiments.
>
> **Response to Weakness 3 and Question 3**
>
> We thank the reviewer for raising this point. Our evaluation already includes MMVP and WildVision, which are explicitly designed to measure visual grounding and vision-centric reasoning. Following prior work[1-5], we report the standard hallucination-focused metrics for these datasets for fair comparison. To further assess generalization beyond hallucination-oriented tasks, we evaluate on SEED-Bench, a widely used general-purpose multimodal benchmark covering reasoning, understanding, and instruction-following questions.
>
> As shown in the table below, MISP-DPO matches or surpasses all baselines across all three backbones, indicating that the gains are not limited to hallucination reduction but transfer to broader multimodal objectives, including general reasoning and instruction following.
>
> | Model Backbone      | Base  | DPO   | mDPO  | CHiP  | Random | **MISP-DPO** |
> |---------------------|-------|-------|-------|-------|--------|--------------|
> | **llava-1.5-7b-hf** | 64.05 | 65.07 | 65.35 | 60.98 | 65.91  | **66.26**    |
> | **Qwen2.5-VL-7B**   | 77.13 | **77.24** | 75.53 | 76.78 | 77.16  | 77.18        |
> | **Qwen2.5-VL-3B**   | 73.99 | 74.02 | 73.20 | 74.10 | 74.25  | **74.32**    |
>
> [1] Wang, Fei, et al. "mDPO: Conditional preference optimization for multimodal large language models." In EMNLP 2024. \
> [2] Fu, Jinlan, et al. "CHiP: Cross-modal hierarchical direct preference optimization for multimodal llms." In ICLR 2025. \
> [3] Fatemeh Pesaran zadeh et al. "LPOI: Listwise Preference Optimization for Vision Language Models." In ACL 2025 Main Conference. \
> [4] Yang, Zhihe, et al. "Mitigating hallucinations in large vision-language models via dpo: On-policy data hold the key." In CVPR 2025. \
> [5] Liu, Wenqi, et al. "Mitigating Hallucination Through Theory-Consistent Symmetric Multimodal Preference Optimization." In NeurIPS 2025.

---

> ### Author Response · Authors · 2025-11-22
> **Response to Reviewer h9st [3/3]**
>
> **Response to Question 2**
>
> We appreciate the reviewer’s question. The computational overhead introduced by the sparse autoencoder and importance-sampling modules is small, as both components are intentionally designed to be lightweight and efficient. Our SAE consists of a single linear encoder and a single linear decoder operating on projected CLIP embeddings, which are low-dimensional and computationally inexpensive. The importance-sampling stage requires only simple vector operations and greedy top-K selection—no backpropagation, no model-dependent scoring, and no additional VLM forward passes.
>
> Crucially, this entire retrieval process is an offline preprocessing step that runs once before DPO training. In our implementation, computing CLIP embeddings takes ~4 minutes, training the SAE takes ~9 minutes, and performing SAE-based negative selection takes ~20 minutes. Only ~33 minutes in total. This corresponds to just **0.37 seconds per image** to select three informative negatives. All components scale linearly with dataset size, operate on fixed-dimension embeddings, and therefore add minimal, predictable overhead. The pipeline introduces no practical bottleneck even when scaling to larger datasets.

---

> ### Author Response · Authors · 2025-11-27
>
> Dear Reviewer h9st,
>
> Thank you for taking the time to review our work and for your constructive comments. We have carefully addressed all of your questions in the rebuttal, including clarifications on the novelty of our multi-negative framework, a detailed analysis of the computational overhead and scalability of the SAE + IS pipeline, and additional results demonstrating generalization tasks.
>
> We hope our responses resolve your concerns. Please feel free to let us know if anything remains unclear. We would be happy to discuss further.
>
> Best regards,
>
> Authors

---

> > ### Comment · Reviewer_h9st · 2025-11-28
> >
> > Thank you for the thorough and constructive review. I appreciate your feedback and will keep my score unchanged.

---

### Official Review · Reviewer_hiMy · 2025-10-31

**Soundness:** 3
**Presentation:** 3
**Contribution:** 3
**Rating:** 6
**Confidence:** 3

**Summary:**

This work introduces MISP-DPO, a novel framework to leverage a sparse auto encoder to identify diverse negative images for multi-negative preference optimization, building on multimodal DPO.
The authors demonstrate the efficacy of their method on multiple multimodal models across a range of benchmarks.

**Strengths:**

* The authors report non-trivial improvements over respective baselines across benchmarks and models. They re-implement DPO, mDPO, and CHiP for a direct and fair comparison (matching data and base models).
* The main contributions are the introduction of multi-negative preference optimization and the method proposed for selecting counterfactual images, building on CLIP retrieval, a sparse autoencoder, and a greedy algorithm to achieve diverse negatives.

**Weaknesses:**

* Section 4.2, describing one of the main contributions of the work, is perhaps a bit limited in detail. For example, the training (data, recipe) for the SAE is not described. And while the math presented in the negative selection may be sufficient, some discussion behind the intuition of sampling for reconstruction error and activation may make the paper more accessible, particularly to casual readers less familiar with using SAEs for interpretability.
* Another recent work [S-VCO] also argues for negative images that are substantially similar to the request image under alignment. In this work, the authors acknowledge this work and argue that this method is expensive (as [S-VCO] relies on image generation method to generate counterfactuals) while the proposed method is more efficient. However, the respective efficacy is not further discussed. An ablation comparing the retrieval + SAE based approach directly to the generative approach proposed by [S-VCO] would further enhance the contributions of this paper. ([S-VCO]’s data (MVC) appears to have been made available.)
* In table 2, the caption implies that the main difference is how negative samples are chosen, but another difference seems to be the number of negative examples being used as per the description in the text. Perhaps this could be clarified?
* For the ablations in table 2, “diffusion” and “crop+diffusion” have two or one negative images still selected by the proposed method as described in the text. I believe this may make comparison a bit more difficult? I understand that multiple negatives based on diffusion or cropping may not have enough diversity, but if that is the concern, perhaps an ablation with 1 negative sample for all methods could be made fairly, further separating the improvements achieved from the targeted selection method from the multi-negative proposal?
* Minor notes for table 2: typo “mdpo” (instead of “mDPO”); Missing average improvement as in table 1 for easier comparisons.


[S-VCO] Wu, Shengguang, et al. "Symmetrical visual contrastive optimization: Aligning vision-language models with minimal contrastive images." arXiv preprint arXiv:2502.13928 (2025).

**Questions:**

* Considering that $d_i$ is dependent on $m_p$, $x$ and $m_n$, how scalable is the retrieval of negatives at training time, if one assumes potentially scaling up the distractor pool and the data used for alignment?
* In the reproduction of mDPO, section 5.4 mentions “mDPO, which relies on a single diffusion-generated negative”. But mDPO constructs the negative image via random cropping (0-20%). Is this a typo?
* It is not clear to me why selecting more than 3 negatives would be detrimental to performance as presented in figure 2 and briefly discussed in 5.4. The authors propose this may be “due to noise introduced by redundant or low-quality samples”, but then redundancy may be directly addressed through the diversity-promoting selection and COCO may not have substantial amounts of “low-quality samples”?
* The reported numbers for MMHalBench for at least LLaVA 1.5 7B seem surprisingly strong, even for reported baselines? Earlier works such as [MDPO] has baseline LLaVA 1.5 7B at 2.19 (in this paper: 2.78) and with their method they achieve “only” 2.39, whereas the “mDPO” reproduction in this paper reports 2.99. Are the evaluation protocols comparable?


[MDPO] Wang, Fei, et al. "mdpo: Conditional preference optimization for multimodal large language models." arXiv preprint arXiv:2406.11839 (2024).

---

> ### Author Response · Authors · 2025-11-21
> **Response to Reviewer hiMy [1/2]**
>
> We thank the reviewer for the detailed and thoughtful comments. We are encouraged by the recognition of our motivation, empirical results, and the relevance of our work. Below, we provide responses to the key concerns and suggestions, and will revise the paper accordingly to improve clarity。
>
> **Response to Weakness 1**
>
> We thank the reviewer for the helpful suggestion and appreciate the opportunity to clarify the SAE training procedure and the intuition behind the signals it provides.
>
> Our SAE is trained offline on contrastive CLIP-based features derived from the RLHF-V dataset together with a large pool of COCO negatives. For each text–image pair, we compute projected CLIP representations and form a contrastive difference vector between the positive and sampled negatives. This fixed-dimensional signal is then modeled by a lightweight SAE with a single linear encoder and decoder. We will add a short description of this setup for clarity.
>
> Regarding the intuition behind reconstruction error and sparse activations, the two signals capture complementary aspects of visual–textual mismatch. Reconstruction error highlights negatives whose differences from the positive image are not well explained by the learned semantic factors, while sparse activations reveal the specific semantic dimensions along which the mismatch occurs. Together, they help identify negatives that are both informative and semantically diverse.
>
> **Response to Weakness 2**
>
> We appreciate the reviewer’s interest in comparing our approach with S-VCO. However, S-VCO operates in a different setting: it generates counterfactual negatives by editing or augmenting the positive image, whereas our method selects negatives directly from an external image pool. Because the data formulation and negative construction pipelines differ substantially, a direct ablation would be difficult to interpret and may not provide a fair or meaningful comparison.
>
> To address the reviewer’s concern about efficiency, we report the computational cost of our full negative-selection pipeline here. On a single A100 GPU, using a 5k RLHF-V dataset together with 118k COCO images, computing CLIP embeddings took 4 minutes, training the SAE took 9 minutes, and negative selection took 20 minutes, 33 minutes in total. This corresponds to only 0.37 seconds per positive sample to select three informative negatives, demonstrating that the overall cost is negligible compared to multimodal DPO training. These results demonstrate that our approach is lightweight and computationally efficient.
>
> **Response to Weakness 3**
>
> In Table 2, all methods use the same number of negative samples per instance except for mDPO; the only difference lies in how these negatives are constructed. The variations mentioned in the text refer to the composition of the three negatives (e.g., diffusion vs. crop+diffusion vs. similarity) rather than a change in their total number. We already clarified this in Section 5.4, where we state that all five strategies share the same model architecture and loss formulation and differ only in how negative images are constructed.
>
> **Response to Weakness 4**
>
> We would like to clarify that Table 2 is evaluated under our **multi-negative DPO setting**, where Eq. 6 defines the image loss as a softmax aggregation over three negatives. In the diffusion and crop+diffusion variants, only the way one or two negatives are constructed is changed, while both the loss function and the total number of negatives remain fixed. This ensures that the comparison is controlled and directly isolates differences in negative-selection strategies. We use three negatives because this is the setting our method is designed for, and changing all methods to a single-negative regime would modify the underlying training dynamics rather than provide a fair ablation. This design also aligns with prior multi-negative studies such as S-DPO, which compare different negative-construction strategies under a fixed negative count [1].
>
> [1] Chen, Yuxin, et al. "On softmax direct preference optimization for recommendation." In NeurIPS 2024.

---

> ### Author Response · Authors · 2025-11-21
> **Response to Reviewer hiMy [2/2]**
>
> **Response to Weakness 5 and Question 2**
>
> We thank the reviewer for pointing out these typos, and we will correct them in the updated version.
>
> **Response to Question 1**
>
> Our retrieval process scales efficiently because all computations operate on fixed-dimensional CLIP embeddings, not raw images. Even though each $d_i$ depends on $(x, m_p, m_n)$, we never perform on-the-fly image retrieval over a large distractor pool. Instead, we precompute CLIP embeddings once for both the alignment data and the global distractor pool, which makes the overall cost linear in the pool size O(N). SAE training and negative selection then operate only on small candidate subsets for each sample, also scaling linearly.
>
> To provide concrete efficiency numbers, we reran the entire retrieval and SAE selection pipeline. On a single A100 GPU, using a 5.7k RLHF-V subset and 118k COCO images, CLIP embedding extraction took 4 minutes, SAE training took 9 minutes, and negative selection took 20 minutes, for a total of 33 minutes, which corresponds to only ~0.37 seconds to select three informative negatives per positive sample.
>
> **Response to Question 3**
>
> We thank the reviewer for raising this point. In practice, we find that adding more than three negatives introduces diminishing returns because the additional retrieved candidates often lie along similar semantic directions. Even with SAE-based diversity promotion, the embedding space of COCO-like natural images contains many near-duplicate or weakly informative negatives once the strongest 2–3 dimensions of disagreement have already been covered. Including more such overlapping negatives effectively amplifies noise in the DPO objective rather than providing new learning signal, which explains the slight performance drop we observe. Importantly, this effect is not due to “low-quality data”, but rather to redundancy in the semantic differences available for a given positive image.
>
> **Response to Question 4**
>
> Thank you for pointing this out. We would like to offer a clarification. The evaluation protocols are aligned, but the absolute MMHalBench scores can differ depending on the GPT judge used for scoring. As noted in our paper, our experiments use GPT-4.1-mini as the evaluator, whereas the mDPO paper appears to use a different GPT model, which naturally leads to shifts in the absolute baselines. Crucially, all comparisons in our study are conducted under a consistent evaluation setup, so the relative performance trends we report remain sound.

---

> ### Author Response · Authors · 2025-11-27
>
> Dear Reviewer hiMy,
>
> Thank you for the thoughtful review and for highlighting the strengths of our framework and empirical results. We have carefully addressed all of your concerns in our rebuttal, including adding clearer details on SAE training and intuition, clarifying the negative-selection setups in Table 2, correcting language issues, and providing expanded explanations regarding scalability and evaluation consistency.
>
> We hope these updates resolve your concerns, and we would be happy to clarify anything further if needed.
>
> Best regards,
>
> Authors

---

### Official Review · Reviewer_HwrY · 2025-11-02

**Soundness:** 2
**Presentation:** 3
**Contribution:** 2
**Rating:** 4
**Confidence:** 4

**Summary:**

The paper proposes MISP-DPO, a listwise preference-optimization framework for multimodal LLMs that combines (i) a Plackett–Luce (PL) objective over a *winner + multiple negative* candidates, (ii) importance sampling with a learned proposal (q_\phi) to reduce training cost and variance, and (iii) a semantically diverse negative-construction pipeline built in CLIP embedding space using sparse autoencoders (SAE) and feature-level perturbations. The method is applied to both *visual-preference* (image conditioned) and *textual-preference* (response conditioned) settings. Experiments report consistent gains over DPO/mDPO/CHiP-style baselines on datasets such as MMHal-Bench, HallusionBench.

**Strengths:**

- **Clear problem framing & practical motivation.** The paper demonstrates clear problem formulation and presentation. The PL-based listwise loss directly optimizes rankings rather than isolated pairs.
- **Good objective with scalable estimation.** Using **importance sampling** and a learned proposal (q_\phi) to approximate the PL gradient is a sound strategy to keep many-negative training tractable while emphasizing informative (“hard”) negatives.
- **Semantically diverse negatives grounded in CLIP space.** The SAE-based feature editing and “mix-and-match” construction pipeline plausibly increases negative diversity without requiring extra human labels.
- **Broad evaluation.** The method is tested across multiple hallucination/factuality benchmarks frequently used for MLLMs (MMHal-Bench, HallusionBench, POPE, WildVision, MMVP) and with a modern evaluation toolkit (VLMEvalKit).

**Weaknesses:**

- **Novelty concern.** Beyond pairwise (1 chosen, 1 rejected) multimodal DPO, there already exists listwise optimization in the context of multimodal DPO, such as LPOI[1]. Therefore, there may be certain novelty concern, **especially when the authors claim** their paper is

  - "the **first** framework to incorporate *multiple*, semantically *diverse* negative images in multimodal DPO" (line 16, in abstract)
  - "the **first** framework to incorporate multi-negative supervision into multimodal DPO" (line 89),
  - "However, such techniques **remain underexplored** in vision-language models" (line 117).

  **So please double-check your claims in the submission.** Also, the use of Plackett–Luce objective is also not really a novelty for listwise DPO, as there are already some prior work such as PLPO[2].

- **Insufficient experiments against former baselines.** As there are already many works published on multimodal DPO, it's not enough to just incorporate mDPO and CHiP as baselines apart from Random and basic DPO in experiments. Please at least incorporate and run experiments for latest methods OPA-DPO[3] and SymMPO[4]. Any more baselines are also welcome.

- **Estimator properties insufficiently analyzed.** The text would benefit from formal statements or empirical diagnostics of *bias/variance* under finite negative sampling, any **weight clipping** or self-normalization, and the stability of (q_\phi) training (e.g., divergence from target leading to high-variance importance weights). (I did not see explicit guarantees/ablation in the provided pages.)

- **Ablation depth.** While the framework has several moving parts (PL listwise loss, IS with (q_\phi), SAE-based negatives, textual-preference branch), the paper would benefit from *systematic ablations* that isolate each contribution and report uncertainty (std/CI over seeds).

References:

[1] Fatemeh Pesaran zadeh et al. "LPOI: Listwise Preference Optimization for Vision Language Models" In ACL 2025 Main Conference.

[2] "Plackett–Luce Preference Optimization (PLPO): Listwise Ranking for Preference Optimization" Preprint 2024.

[3] Yang et al. "Mitigating Hallucinations in Large Vision-Language Models via DPO: On-Policy Data Hold the Key" In CVPR 2025.

[4] Liu et al. "Mitigating Hallucination Through Theory-Consistent Symmetric Multimodal Preference Optimization" In NeurIPS 2025.

**Questions:**

1. **Cost accounting:** Please add wall-clock/GPU-hour comparisons vs. DPO, mDPO, CHiP to demonstrate the promised efficiency gains of importance sampling.
2. **Negative construction controls:** How do you ensure that CLIP-SAE-driven negatives are not trivially separable (e.g., distributional artifacts), and that they **stress visual grounding** rather than language priors? Any human spot-checks?
3. **Unbiasedness & variance:** Is the IS gradient strictly unbiased under your training scheme? Do you apply **weight clipping** or **self-normalized IS**? Please report effective sample sizes or variance diagnostics across training.

---

> ### Author Response · Authors · 2025-11-21
> **Response to Reviewer HwrY [1/4]**
>
> We thank the reviewer for the thorough and thoughtful evaluation. We address the raised concerns below and clarify how our work handles them, with references to the relevant sections of the paper.
>
> **Response to Weakness 1**
>
> We thank the reviewer for pointing out the related work. Our paper addresses a different and currently unaddressed challenge: constructing and integrating multiple, semantically rich, multifaceted negative images from open-domain sources. Unlike prior work, our contribution is not merely using a listwise objective, but discovering and leveraging orthogonal visual deviations via SAE-guided importance sampling, and then coupling them with a multimodal Plackett–Luce objective.
>
> Regarding related work, LPOI generates negative images by perturbing a single positive image, producing a sequence of homologous variants with limited semantic diversity. In contrast, our negatives originate from open-domain image pools and are explicitly selected to be both multiple and semantically diverse. PLPO employs a Plackett–Luce objective but focuses entirely on the text domain and does not address the visual-negative construction problem central to multimodal alignment.
>
> Our paper explicitly emphasizes two components:
> (1) the use of multiple negatives, and
> (2) ensuring these negatives are semantically diverse, not just multiple variants of the same seed image.
> As stated in the abstract (line 16) and introduction (line 53), our novelty lies in being the first to introduce multi-negative and semantically diverse supervision into multimodal DPO. To avoid ambiguity, we will improve the wording for clarity.
>
> **Response to Weakness 2**
>
> | Method    | MMHalBench Score ↑ | MMHalBench HalRate ↓ | HallusionBench aA ↑ | HallusionBench fA ↑ | HallusionBench qA ↑ | POPE Acc ↑ |
> |-----------|---------------------|-----------------------|----------------------|----------------------|----------------------|-------------|
> | OPA-DPO   | 3.51                | 33.33                 | 52.05               | 21.09               | 19.56               | 82.64       |
> | SymMPO    | **3.61**            | **32.29**             | 57.09               | 24.56               | 23.95               | 81.25       |
> | MISPDPO  | 3.51                | **32.29**             | **57.52**           | **25.43**           | **24.83**           | **83.94**   |
>
> We thank the reviewer for the suggestion and have added experiments with OPA-DPO and SymMPO. As shown in the updated table, MISP-DPO remains the strongest overall, achieving the best performance on all HallusionBench metrics (57.52/25.43/24.83) and POPE accuracy (83.94), which directly measure visual hallucination. Although SymMPO attains a slightly higher MMHalBench score, both OPA-DPO and SymMPO mainly emphasize text-side preference optimization and rely on very simple image negatives (cropping or light perturbations). In contrast, our method focuses on constructing semantically diverse, multi-facet image negatives, which is crucial because VLMs consistently struggle more with image-grounded reasoning than text[1-3]. This difference explains why MISP-DPO provides stronger gains on hallucination benchmarks. Importantly, our approach is complementary to both baselines. Their methods do not address diverse image-negative construction, and our SAE-guided selection can be directly combined with them. We will include these results in the revised paper.
>
> [1] Wang, Fei, et al. "mDPO: Conditional preference optimization for multimodal large language models." In EMNLP 2024 \
> [2] Fu, Jinlan, et al. "CHiP: Cross-modal hierarchical direct preference optimization for multimodal LLMs." In ICLR 2025. \
> [3] Wu, Shengguang, et al. "Symmetrical visual contrastive optimization: Aligning vision-language models with minimal contrastive images." arXiv preprint arXiv:2502.13928 (2025).

---

> ### Author Response · Authors · 2025-11-21
> **Response to Reviewer HwrY [2/4]**
>
> **Response to Weakness 3 and Question 3**
>
> In our implementation, we do not apply weight clipping or self-normalized IS; instead, we rely on (i) gradient clipping to bound $\|\Delta_{\theta}\|$ and (ii) an SAE-guided sampler $q_{\phi}$ that closely tracks $p_{\theta}$ and avoids overly peaked importance weights. Below, we provide formal statements and proofs showing that our IS gradient is strictly unbiased and that its variance admits a finite-sample bound. We have also incorporated these theoretical results into the revised paper.
>
> For convenience, we define the inner gradient term in Lemma 4.1,
> \begin{equation}
>     g(\theta)
>     := \sum_{i=1}^N p_\theta(m_n^i \mid x, m_p, y_p)\
>     \Delta_\theta(m_n^i, m_p \mid x, y_p).
> \end{equation}
>
> Our importance-sampling scheme (Sec. 4.1) introduces a proposal distribution
> $q_\phi(m_n \mid x, m_p, y_p)$ induced by the CLIP+SAE pipeline, and samples a small candidate pool $\tilde{\mathcal{S}}_n$. We then approximate the exact gradient using an importance-weighted estimator.
>
> Let the importance weight be
> \begin{equation}
>     w(m_n) = \frac{p_\theta(m_n \mid x, m_p, y_p)}{q_\phi(m_n \mid x, m_p, y_p)}.
> \end{equation}
>
> **Unbiased Importance Sampling Gradient**
>
> Assume that $q_\phi(m_n \mid x, m_p, y_p) > 0$ whenever $p_\theta(m_n \mid x, m_p, y_p) > 0$.
> Define the Monte Carlo estimator of $g(\theta)$ as,
> \begin{equation}
>     \hat{g_k(\theta)} = \frac{1}{K} \sum_{k=1}^K
>     w\big(m_n^k \big) \Delta_\theta\big(m_n^k, m_p \mid x, y_p\big),\quad
>     m_n^k \sim q_\phi(\cdot \mid x, m_p, y_p).
> \end{equation}
>
> Then $\hat{g_k(\theta)}$ is an unbiased estimator of $g(\theta)$,
> \begin{equation}
>    E_{\{m_n^k\}\sim q_\phi}
>     \big[ \hat{g_k(\theta)} \big]
>     = g(\theta).
> \end{equation}
>
> We updated the detailed proof in the appendix.
>
> Thus, the IS estimator recovers the exact gradient term in expectation, and hence the overall gradient in Eq.(7),
> \begin{equation}
> \nabla_\theta L_{\mathrm{img}}(\theta; \mathcal{S_n})
>     = \beta \sigma\Big(\log \sum_{i=1}^N \exp(a_i)\Big) g(\theta),
> \end{equation}
>
> admits an unbiased importance-sampling estimator by replacing $g(\theta)$ with $\widehat{g}_k(\theta)$.
>
> **Variance Bound Under Finite Negative Sampling**
>
> We next calculate the variance of $\widehat{g}_k(\theta)$ when only a finite number $K$ of negatives are sampled.
>
> Define the maximum importance weight,
> \begin{equation}
>     \|w_\infty\|
>     := \max_{m_n \in \mathcal{S_n}}
>     \frac{p_\theta(m_n \mid x, m_p, y_p)}{q_\phi(m_n \mid x, m_p, y_p)}.
> \end{equation}
> Assume that the gradient differences are bounded,  which in practice is achieved by gradient clipping [4-6]. There exists $L < \infty$ such that,
> \begin{equation}
>     \big\|\Delta_\theta(m_n, m_p \mid x, y_p)\big\|_2 \le L
>     \quad \forall m_n \in \mathcal{S_n}.
> \end{equation}
>
> **Variance Bound**
>
> Under the assumptions above, the mean-square error of the IS gradient estimator is bounded as,
> \begin{equation}
>     E\big[\|\hat{g}_k(\theta) - g(\theta)\|_2^2\big] \le  \frac{L^2}{K}\|w\|.
> \end{equation}
>
> Consequently, the expected absolute deviation satisfies,
> \begin{equation}
>     E\big[\|\hat{g}_k(\theta) - g(\theta)\|_2\big] \le  L\sqrt{\frac{\|w\|}{K}},
> \end{equation}
>
> where $\|w\|$ represents $\|w\|_\infty$. We updated the detailed proof in the appendix.
>
> In our setting, the SAE-guided sampler is explicitly designed to approximate the target distribution with diverse but not overly peaky weights, which empirically keeps $\|w\|_\infty$ small and leads to stable optimization.
>
> [4] Zhang, Jingzhao, et al. "Why gradient clipping accelerates training: A theoretical justification for adaptivity." In ICLR 2020. \
> [5] Menon, Aditya Krishna, et al. "Can gradient clipping mitigate label noise?" In ICLR 2020. \
> [6] Long, Ouyang, et al. "Training language models to follow instructions with human feedback." In NeurIPS 2022.

---

> ### Author Response · Authors · 2025-11-21
> **Response to Reviewer HwrY [3/4]**
>
> **Response to Weakness 4**
>
> | Method         | HallusionBench aA ↑ | HallusionBench fA ↑ | HallusionBench qA ↑ | POPE Acc ↑ | MMVP Acc ↑ |
> |----------------|----------------------|----------------------|----------------------|-------------|-------------|
> | w/o PL loss    | 64.34               | 36.64               | 37.05               | 87.37       | 72.33       |
> | w/o SAE neg    | 64.98               | **38.44**           | 37.36               | 87.52       | 72.00       |
> | Isolate Text   | 63.45               | 34.08               | 35.27               | 85.43       | 71.66       |
> | Isolate Image  | 61.59               | 32.11               | 35.44               | 85.01       | 71.30       |
> | **MISP-DPO**   | **65.51**           | **38.44**           | **38.02**           | **87.77**   | **74.25**   |
>
> Thank you for the suggestion. We have added ablations isolating each major component of our method: (1) removing the PL-based loss (replacing it with a simple average loss), (2) replacing SAE-selected negatives with random COCO negatives, and (3) training with only image-side or only text-side loss. As shown in the updated table, removing either the PL loss or our SAE-guided negative construction consistently degrades performance across all benchmarks, confirming that both informative, semantically diverse negatives and our proposed loss design are essential. Furthermore, using only image loss or only text loss leads to the largest performance drop, indicating that both modalities contribute crucially to effective multimodal alignment.
>
> Could the reviewer clarify what is meant by “isolating IS with $q_\phi$”? We are happy to include an additional ablation once we understand the intended setup. We will add all ablation results to the revised paper.
>
> **Response to Question 1**
>
> We want to clarify that our efficiency claim refers to the negative construction stage, where naïvely evaluating all 118k COCO images would be prohibitively expensive. Our SAE-guided importance sampling selects three high-quality, multi-facet negatives using only CLIP features and a 1-layer SAE, requiring only 0.37 seconds per positive sample to select the three informative negatives. The entire process (SAE training + negative selection) completes in 33 minutes on a single A100, making it substantially more efficient than any full-dataset retrieval.
>
> Regarding training cost, using three negatives naturally introduces more computation compared to baselines that use only one. In practice, the overhead is modest. Text-only DPO trains in ~2.6s per sample, mDPO (with a single perturbed negative) takes ~3.1s, and CHiP exhibits a similar runtime to ours. MISP-DPO completes in ~3.6s. Thus, the training cost increases in a predictable and manageable way, given the clear gains in multimodal grounding.

---

> ### Author Response · Authors · 2025-11-21
> **Response to Reviewer HwrY [4/4]**
>
> **Response to Question 2**
>
> We have already included illustrative examples in Figure 3, which directly demonstrate that our selected negatives are semantically aligned yet visually varied—e.g., all remain within the same scene category (cakes/desserts; desks/computer setups) while differing along meaningful visual facets such as composition, background, object attributes, and viewpoint. The negatives are related to the positive image but differ in ways that require image-grounded reasoning, not textual shortcuts, to distinguish.
>
> Our CLIP–SAE pipeline operates on the cross-product representation of the image and its paired text, ensuring that negative selection is grounded in the visual–semantic relationship rather than in language alone. The SAE identifies latent directions capturing meaningful visual deviations, which prevents trivial separability and encourages negatives that stress image-grounded reasoning (e.g., distinguishing a fruit cake from other cakes).
>
> We also performed light human spot-checks and confirmed that the selected negatives are not random images but remain semantically connected while differing along multiple meaningful visual angles.

---

> ### Author Response · Authors · 2025-11-27
>
> Dear Reviewer HwrY,
>
> Thank you again for the thoughtful and detailed review. We have carefully addressed all of your questions in the rebuttal, including (i) clarifying the novelty of our multi-negative, semantically diverse construction pipeline, (ii) adding the requested baselines (OPA-DPO, SymMPO), (iii) providing formal unbiasedness and variance guarantees for our importance-sampling estimator with full proofs in the appendix, (iv) expanding ablations across all major components, and (v) reporting the detailed cost-accounting analysis.
>
> We sincerely hope that our updates have resolved your concerns. Please let us know if any part of our response remains unclear. We would be more than happy to provide further clarification.
>
> Best regards,
>
> Authors

---

> ### Comment · Reviewer_HwrY · 2025-11-27
> **Response to the Authors**
>
> Thanks for the authors' responses. I think the responses have solved my major concerns.

---

### Author Response · Authors · 2025-12-02
**Summary of Response**

We sincerely thank the Area Chairs for evaluating our submission. We also appreciate the Reviewers HwrY, hiMy, and h9st for their constructive feedback. Below, we (1) summarize the core contributions and significance of MISP-DPO, along with the strengths recognized by the reviewers, and (2) concisely recap the key clarifications and additions made during the rebuttal.

## Highlights of Contributions

1. **Multi-negative, semantically diverse multimodal DPO.**
MISP-DPO introduces the first **multi-negative, semantically diverse** preference optimization framework for multimodal DPO, combining a Plackett–Luce objective with a CLIP + sparse autoencoder guided importance-sampling pipeline that discovers orthogonal visual deviations (object, layout, attributes, context).
Especially, **Reviewer h9st** described this as a principled and well-motivated solution to the single-negative limitation, and **Reviewer hiMy** recognized our multi-negative design and SAE-based construction as the core methodological advance.

2. **Lightweight, theoretically grounded design.**
Our CLIP+SAE negative selection is a **one-time offline process**, introduces only a modest training-time overhead, and now includes formal guarantees of unbiasedness and bounded variance for the importance-sampling estimator.

3. **Convincing Empirical Evaluation.**
We evaluate across multiple backbones and diverse grounding and hallucination benchmarks, MMHal-Bench, HallusionBench, POPE, MMVP, WildVision, where MISP-DPO achieves consistent improvements **(Reviewer HwrY)**. **Reviewers hiMy and h9st** also appreciated the fair re-implementation of DPO/mDPO/CHiP and the consistency of gains across tasks.

4. **Generalization Beyond Hallucination.**
Our improvements extend to general multimodal reasoning and instruction-following (e.g., SEED-Bench), confirming that multi-facet negative supervision benefits broader alignment—a direction reviewers found important and promising.

## Summary of Rebuttal Clarifications

1. **Novelty vs. prior listwise and multimodal DPO methods.**
   We emphasize that our contribution is not simply using a listwise PL loss, but how we construct and integrate **multiple semantically diverse negative images**. Our method is **the first to combine**:

   - SAE-guided semantic decomposition to identify diverse, orthogonal visual deviations from open-domain pools,
   - importance-sampled multi-negative construction that avoids homologous perturbations and scales efficiently, and
   - a multimodal PL objective that integrates these negatives.

   We also added new baselines (OPA-DPO, SymMPO), where MISP-DPO achieves stronger hallucination reduction, strengthening the novelty and necessity of semantically diverse negative selection.

2. **Unbiasedness and variance of the IS estimator (Reviewers HwrY, h9st).**
We added formal statements and proofs showing that: (i) our importance-sampling estimator is strictly unbiased for the PL gradient term, and (ii) the mean-squared error and expected deviation of the IS gradient estimator admit finite-sample bounds. These results are now included in the appendix and directly address the questions about unbiasedness, variance control, and stability.

3. **Ablations and SAE details.**


   - To address **Reviewer HwrY’s** concern about ablation depth, we added component-wise ablations: removing the PL loss, training with image-only or text-only objectives, and replacing SAE-selected negatives with the random-negative variant already shown in Table 1. All of these lead to consistent performance drops, confirming that both PL training and SAE-guided diverse negatives, as well as joint multimodal supervision, are crucial.
   - We clarified the SAE training: the SAE is trained offline to capture meaningful semantic shifts in CLIP space, using reconstruction error and sparse activations to extract informative, diverse deviations. We further included light human spot-checks showing that selected negatives remain semantically related yet visually distinct, addressing **Reviewer hiMy’s** concern.

4. **Efficiency, scalability, and generalization.**


    - We provided concrete runtime evidence showing that the CLIP+SAE selection process is fast, linear-time, and fully offline, requiring only ~0.37 seconds per example to select three informative negatives. Training-time overhead remains modest, making the full pipeline practical at scale.
    - To answer **Reviewer h9st’s** concern about generalization, we further added SEED-Bench results, showing that improvements extend to general multimodal reasoning, not only hallucination tasks.

In summary, we believe we have addressed all concerns raised by the reviewers. Given the novel multi-negative framework, the combination of solid theoretical and empirical results, and the reviewers’ positive and strengthened assessments after the rebuttal, we believe MISP-DPO makes a meaningful and impactful contribution.
Thanks again for your time!

---

### Meta-Review · Area_Chair_FbKQ · 2026-01-07

**Summary:**

MISP-DPO is a listwise preference-optimization framework designed to improve multimodal LLM alignment and reduce hallucinations. Moving beyond standard pairwise DPO, it employs a Plackett–Luce objective to rank a single positive response against multiple diverse negatives.

To ensure these negatives are semantically meaningful, the authors use Sparse Autoencoders (SAEs) in CLIP embedding space to generate diverse feature-level perturbations. The framework further incorporates importance sampling with a learned proposal distribution to maintain training efficiency and reduce variance. Evaluated on benchmarks like MMHal-Bench and HallusionBench, MISP-DPO consistently outperforms existing baselines in both visual and textual preference tasks.

The paper presents a principled and empirically effective extension of multimodal DPO; while the scope of evaluation remains somewhat centered on hallucination tasks, the authors demonstrated a certain level of generalizability to general tasks.

**Reviewer Concerns:**

The core strength of this paper lies in its well-formulated approach to multimodal preference optimization. By combining a Plackett–Luce objective with SAE-guided importance sampling, the authors effectively leverage orthogonal visual deviations to improve model alignment. The reviewers found the problem framing to be both practical and well-motivated, noting that the experiments demonstrate consistent, non-trivial performance gains across the selected benchmarks.

Initial concerns were raised regarding the authors' claims of being the "first" to explore listwise multimodal optimization, with reviewers pointing to existing works such as LPOI and PLPO. Additionally, Reviewer HwrY noted a lack of comparisons against contemporary baselines and a need for deeper ablation studies.

Through the rebuttal process, the authors successfully addressed these points by:
* Clarifying and tempering their novelty claims relative to prior work.
* Incorporating additional baseline comparisons and ablation experiments.

As a result, Reviewers HwrY and h9st acknowledged that their primary concerns regarding technical depth and prior art have been resolved.

The primary outstanding concern involves the generalizability of the framework beyond hallucination-focused tasks. While the authors added results from SEED-Bench during the rebuttal, MLLMs are increasingly held to a broader standard across diverse capabilities. To fully demonstrate the robustness of MISP-DPO, evaluation on a wider array of benchmarks—such as MME, MM-Vet, or MMMU—would be beneficial to verify performance in text-rich, knowledge-heavy, and grounding-based scenarios.

**Reviewer Scores:**

Reviewer HwrY commented the major concern has been addressed but didn't raise the score.

Reviewer hiMy is likely to remain his positive score if all concerns have been addressed.

Reviewer h9st acknoledged the response and remains the positive score.

---

### Decision · Program_Chairs · 2026-01-26

Accept (Poster)